

# Regional index flood estimation at multiple durations with generalized additive models

Danielle M. Barna[1,3], Kolbjørn Engeland[1], Thomas Kneib[5], Thordis L. Thorarinsdottir[2,4], and
Chong-Yu Xu[3]

[1]Norwegian Water Resources and Energy Directorate, P.O. Box 5091 Majorstua, NO-0301 Oslo, Norway.
[2]Norwegian Computing Centre, P.O. Box 114 Blindern, NO-0314 Oslo, Norway
[3]Department of Geosciences, University of Oslo, P.O. Box 1047 Blindern, NO-0316 Oslo, Norway.
[4]Department of Mathematics, University of Oslo, Oslo, Norway.
[5]Chair of Statistics, University of Göttingen, Göttingen, Germany.

**Correspondence:** Danielle M. Barna (daba@nve.no)

**Abstract.** Estimation of flood quantiles at ungauged basins is often achieved through regression based methods. In situations where flood retention is important, e.g. floodplain management and reservoir design, flood quantile estimates are often needed at multiple durations. This poses a problem for regression-based models as the form of the functional relationship between catchment descriptors and the response may not be constant across different durations. A particular type of regression model that is well-suited to this situation is a generalized additive model (GAM), which allows for flexible, semi-parametric modeling and visualization of the relationship between predictors and the response. However, in practice, selecting predictors for such a flexible model can be challenging, particularly given the characteristics of available catchment descriptor datasets. We employ a machine learning-based variable pre-selection tool which, when combined with domain knowledge, enhances the practicality of constructing GAMs. In this study, we develop a GAM for index (median) flood estimation with the primary objective of investigating duration-specific differences in how catchment descriptors influence the median flood. As the accuracy of this explainable approach is dependent on the fitted GAM being adequate, the secondary objective of our study is prediction of the median flood at ungauged locations and multiple durations, where predictive performance and reliability at ungauged locations are used as proxies for adequacy of the GAM. Predictive performance of the GAM is compared to two benchmark models: the existing log-linear model for median flood estimation in Norway and a fully data-driven machine learning model (an extreme gradient boosting tree ensemble, XGBoost). We find that the predictive accuracy and reliability of the GAM matched or exceeded that of the benchmark models at both durations studied. Within the predictor set selected for this study, we observe duration-specific differences in the relationship between the median flood and the two catchment descriptors effective lake percentage and catchment shape. Ignoring these differences results in a statistically significant decline in predictive performance. This suggests that models developed and estimated for prediction of the index flood at one duration may have reduced performance when applied directly to situations outside of that specific duration.





# 1 Introduction

Estimating flood quantiles in ungauged catchments is a frequent challenge in hydrology and is crucial to many tasks related to design of infrastructure, emergency and land use planning. A common method for flood quantile estimation at ungauged catchments is regional flood frequency analysis (RFFA), which uses catchment characteristics–that is, descriptors of the physical properties of the catchment such as, for example, area, lake percentage, average elevation and total river length–as well as climate characteristics like precipitation and temperature to infer flood quantiles at a specified ungauged location. Often this inference is based on the concept that spatial variations in flood statistics are closely linked with regional catchment and climate characteristics and is achieved through regression based methods (Robson and Reed, 1999).

This study focuses on constructing regression models for the median annual maximum (index) flood at multiple flood durations. Regression models on flood quantiles are typically constructed for a single flood duration. However, for many hydrologic applications where flood retention is important, e.g. floodplain management and reservoir design, flood quantile estimates for different durations are needed. Regression models used for index flood estimation are typically *parametric* regression models (e.g. linear, log-linear, nonlinear, or generalized linear models). These models rely on a parametric description of what is called the *functional form* between predictors and response. That is, we assume the relationship between the median flood and the mean temperature in February, for example, is completely described by the functional form $x^2$. In this situation, we would need to estimate a regression coefficient—that is, we would need to estimate the magnitude and direction of this functional form—but the underlying relationship will always be described by the square function. Parametric models are easy to interpret and estimate and therefore widely used. However, in the situation where index flood estimates at multiple durations are needed, use of the same parametric model at each duration assumes that, although the magnitude and direction of the functional form may change when regression coefficients are re-estimated or scaled for different durations, the relationship between catchment descriptors and the index flood will always be described by the same functional form regardless of the duration being considered. This is a common assumption among models that seek to provide index flood estimates at different durations, for example regional flood-duration-frequency (QDF) models as in Javelle et al. (2002). Here the regional QDF model is an extension of the standard index flood approach and includes an additional "characteristic duration" parameter that acts as a scaling factor on the coefficients of a parametric index flood regression model. This helps enforce consistency between flood quantile estimates at different durations; however, the underlying assumption is not investigated.

Overall, there is a gap in regional flood frequency analysis when it comes to assessing regression models for flood quantiles at multiple durations, especially considering the possibility that the functional form describing the relationships between catchment descriptors and flood quantiles may not remain constant across durations. In this study we address duration-specific differences in regional index flood estimation for applications where the total volume of water is of interest. We consider two different flood durations: 1 and 24 hours. The focus on flood-retention specific applications means the durations in this study represent the total flow volume over 1 and 24 hours, not flood events that lasted for precisely 1 or 24 hours. Annual maxima corresponding to the these durations are generated by sampling from the discharge series averaged over the desired time window.



The underlying relationship between flood quantiles and catchment descriptors is likely to be nonlinear (Pandey and Nguyen, 1999; Tarquis et al., 2011) as the underlying hydrological processes are non-linear in nature (Durocher et al., 2015). Some reasons for the non-linear nature are that the runoff response to rain and snow melt is non-linear (Gioia et al., 2012), snow melt processes are non-linear, and the key flood generating processes might depend on events as well as catchments. The classic way of handling this nonlinearity is to transform the predictors such that they have a linear relationship with flood

quantile and fit a linear or log-linear regression model. This is the approach used in the current regional median flood model for Norway (Engeland et al., 2020). In this approach, much of the work in defining an appropriate model lies in finding the suitable polynomial terms and transformations of predictors to enhance the fit of the linear relationship. This can be a laborious and inaccurate process. In situations where model interpretation and uncertainty analysis are priorities, an appealing alternative to linear or log-linear models are generalized additive models (GAMs).

GAMs are semi-parametric extensions of the linear regression model that can account for nonlinear relationships between predictors and response. GAMs are often referred to as "data-driven", meaning the data determine the form of the relationship between the response and the predictors rather than assuming some form of parametric relationship, e.g. transformation of predictors. The main use case of GAMs lies in applications where a nonlinear relationship between the predictor and response needs to be defined or established; however, if the relationship is linear, the smooth function that defines the relationship

between predictor and response will recover the linear relationship. This offers a potential simplification of the modeling process as we no longer need to identify appropriate predictor transformations. Furthermore, the fact we no longer have to specify a functional form of the predictor-response relationship has potential to generate relationships that better represent the underlying data. This allows for statistical analyses that focus on identification and description of the data-driven relationships between predictor and response. These descriptive statistical analyses are a valuable tool for addressing model assumptions.

Additionally, the relationships identified can in some cases increase our understanding of hydrologic systems, although the reality of the functional relationships should always be established by theory or process-based models outside of the statistical analysis.

Recent years have seen increasing use of GAMs in hydrology, often in situations where the relationship between predictor and response variables is complex and nonlinear, but the available data limits the application of full-scale machine learning

models. In many cases the uncertainty and reliability assessments offered by GAMs are important. Some examples using GAMs include reconstruction of reservoir operation signals (Brunner and Naveau, 2023), forecasting spring flood peaks (Dubos et al., 2022), predicting decadal statistics of daily streamflows (Crowley-Ornelas et al., 2023), and forecasting drought conditions (Mathivha et al., 2020). Specific application of GAMs for estimation of flood quantiles is generally first attributed to Chebana et al. (2014). Here GAMs were used to estimate the quantiles corresponding to the 10, 50 and 100 year return periods and

the models were compared to log-linear models on a variety of different regional groupings. The GAMs were found to have improved predictive performance and the flexibility of the GAMs reduced the need to split into hydrologically homogenous regions. Other examples of GAMs used for flood quantile regression are the comparative studies of Msilini et al. (2022) and Rahman et al. (2018), which compared GAMs to traditional log-linear regression approaches, among others. Rahman et al. (2018), in line with Chebana et al. (2014), found that GAMs typically outperformed log-linear models, even without





constraining the GAM to hydrologically similar neighborhoods or regions of influence. In an application that looked at quantile-specific performance differences, Noor et al. (2022) compared a GAM to a linear quantile regression technique and found the GAM resulted in improved performance, but only on quantiles associated with small return periods of up to ten years.

Since GAMs are so flexible, selection of the most relevant predictors must be undertaken carefully to avoid overfitting and obtain robust predictive models. The natural approach to variable selection within GAMs are the shrinkage-based methods

developed by Marra and Wood (2011). These methods involve shrinking the smooth function estimates of the GAM towards zero, such that the relative contribution of each parameter reflects its importance. This approach offers the benefits of subset selection while allowing variable selection to be accomplished in a single step, and is particularly appealing as it allows for variable selection uncertainty to be included in the final model. However, the usefulness of these shrinkage methods is limited to relatively small sets of uncorrelated variables. This is problematic for our application as typical hydrologic data

sets used in regional analyses contain a large number of candidate variables, many of which are highly correlated. For these reasons, current applications of GAMs to flood quantiles commonly rely on backwards stepwise selection (Chebana et al., 2014; Rahman et al., 2018; Noor et al., 2022; Msilini et al., 2022) sometimes coupled with a pre-selection step as in Dubos et al. (2022). Backward selection approach used for GAMs has the potential to select appropriate covariates at the same rate as shrinkage approaches, but only when the information content of the data is high (Marra and Wood, 2011). Stepwise procedures

are also prone to well-known problems–namely inconsistency in the selected variable sets and inability to account for variable selection uncertainty–most of which were the motivators for developing the theory around shrinkage estimators for GAMs (Marra and Wood, 2011).

In scenarios where limited information is available for informed predictor variable choice, an idea to increase the practicality of using shrinkage estimators is to use a machine learning model to aid in selection of a small, nonredundant set of predictors

that can then be validated with shrinkage based methods in GAMs. Guisan et al. (2002) highlights the potential of machine learning techniques to complement GAMs by uncovering nonlinear, and possibly previously unknown, relationships between predictors and the response variable. In this complementary approach, the machine learning model is a practical tool used in conjunction with expert judgement to aid in initial predictor variable selection. Use of this "tool" occurs prior to the construction of the GAM and can be fully replaced with expert judgement in situations where the potential set of appropriate predictors

is well-defined. There exist a wide variety of machine learning-based algorithms for predictor selection; for the classic introduction to the different models and taxonomies available see Guyon and Elisseeff (2003). We chose the tree-based Iterative Input Selection algorithm (IIS) presented in Galelli and Castelletti (2013) and applied in, for example, Prasad et al. (2017), He et al. (2022) and Pesantez et al. (2020). The algorithm was developed for application to hydrology, contains routines for selecting nonredundant predictors, and provides an accessible way to limit variable interactions. This last point in particular is

important to our application as variable interactions are not considered in the GAM presented here.

This study develops a GAM for estimation of the median annual maximum flood. The primary objective of our study is detection and description of the functional relationships between the median flood and catchment covariates at both the 1 hour and 24 hour durations. Here we assume that, while the relationship between the covariates and the median flood may vary with duration, the covariates themselves remain constant across different durations. The accuracy of this explainable approach



is dependent on the fitted GAM being adequate. Thus the secondary objective of our study is prediction of the median flood
at ungauged locations, where predictive performance and reliability at ungauged locations are used as proxies for adequacy
of the GAM. We use two benchmark models to establish predictive performance. These are the existing log-linear model
for median flood estimation in Norway and a gradient-boosted tree ensemble (XGBoost). XGBoost has established use in
hydrology (Zounemat-Kermani et al., 2021) and is applied in, for example, Laimighofer et al. (2022a) and Ni et al. (2020). As

part of what distinguishes the GAM from the log-linear model is the flexible, data-driven nature of the response relationship,
it is useful to have a comparison point from a fully data-driven model such as XGBoost. The following research questions will
be addressed: (i) Can the GAM achieve comparable or improved performance compared to the benchmark models on the 1
hour and/or the 24 hour duration? (ii) Can we identify and describe duration-specific differences in how catchment covariates
influence the median flood? How impactful are these differences? (i.e. if we ignore them, what is the impact on predictive

performance?). Our analysis will be performed on annual maximum data since flood guidelines in Norway pertain to annual
maximum values.

    The remainder of the paper is organized as follows: section 2 introduces the flood data and catchment descriptors. Section
3 presents an outline of the study design. Section 4 presents the GAM used in this study and summarizes the chosen predictor
selection approach. This section also summarizes the two reference models and the evaluation methods used to assess all

models in the study. The results section 5 presents the predictive performance and model reliability results as well as the
functional relationships identified by the GAM. The paper finishes with a discussion (section 6) and conclusions (section 7).

## 2  Data

Flood data from 232 gauging stations across Norway were used in this study (Fig. 1). The stations exhibit a diversity of hydro-
climatic regimes relative to Nordic catchments. The spatial distribution of temperature and precipitation regimes in Nordic

countries is primarily influenced by climatological gradients associated with latitude, topography, and proximity to the coastal
zone; the diverse topography and wide range of latitudes in Norway make it a suitable location for regionalization studies in
the Nordic region.

    In Norway the two major flood generating processes are snowmelt and rainfall. The regional importance of snowmelt as a
runoff generating process varies greatly due to differences in the temperature regime, snowpack volumes and the snow season

across the country. Inland and northern regions are those primarily driven by snowmelt and experience prominent high flows
during spring and summer, while western and coastal regions are primarily driven by rainfall and experience high flows during
autumn and winter. However, local climate and mixed or transitional flood regimes mean these regional patterns exhibit great
variability, and seasonal patterns are not very distinct in rainfall-driven catchments (Vormoor et al., 2016).

    The observed streamflow time series were obtained from the national hydrological database Hydra II hosted by the Nor-

wegian Water Resources and Energy Directorate (NVE). The streamflow records have at least 20 years of quality controlled
data for periods with minimal influence from river regulations and a sufficient quality for high streamflows; see Engeland et al.
(2016) for details.





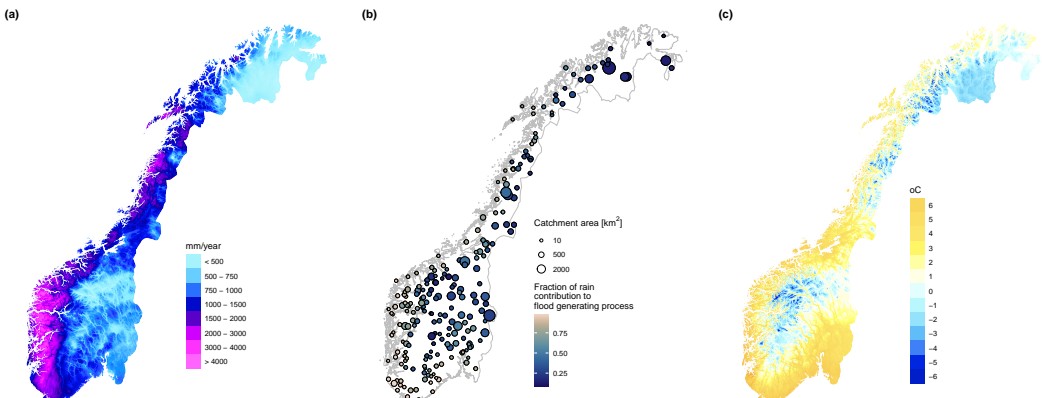

**Figure 1.** Panel (a) shows average rainfall totals (mm) for the entire year from the period 1991-2020. Panel (b) shows locations of the 232 gauging stations used in this study, where catchment area and average fraction of rain contribution to flood are indicated by size and color, respectively. Panel (c) shows average temperature (°C) for the entire year from the period 1991-2020.

The flood data used in this study is the median annual maximum flood [l/s/km2] at each station. We used both the 1-hour and 24-hour median annual maximum flood. The data at different durations were constructed via an aggregation-based approach, where the durations represent the total volume of water that arrives over a time span of 1 and 24 hours, not flood events that lasted precisely 1 or 24 hours. This approach is used in, for example, Breinl et al. (2021) and Barna et al. (2023). For each station, even spacing in the streamflow time series was enforced via regular sampling of a linear interpolation of the observed data. A moving average with a window length of either 1 or 24 hours was then applied to the evenly spaced streamflow time series. From the smoothed time series, annual maxima were extracted to create separate sets of maxima for the 1-hour and 24-hour durations. We then computed the median of these sets of maxima to get the 1-hour and 24-hour median annual maximum flood at each of the 232 stations.

## 2.1 Data quality control

Given the focus on sub-daily floods, it is necessary to make sure that the sampling frequency of the data is high enough to represent peak flood magnitudes with sufficient quality. Each of the streamflow records contains a variety of collection methods. These differing collection methods provide data at different frequencies. Generally, the earlier part of the streamflow record has daily time resolution, while the later part of the record contains a higher frequency of measurements after adoption of digitized limnigraph records and/or digital measurements. For our dataset, the shift to a higher frequency of measurements is typically around 1980, and stations have, on average, 27 years of high frequency data. The time resolution of the digital measurements and the digitization of the limnigraph records were selected by NVE to be frequent enough to represent flood peaks at individual stations. Total record lengths in our data set range from a minimum of 20 years of data to 129 years at station 62.5 (Bulken); the distribution of total record lengths is plotted in Fig. 2.





The median annual maximum flood at both the 1- and 24-hour durations is computed over the total number of years of data available at each station. This means that for certain stations, especially those with longer record lengths, the median is constructed from annual maxima derived from streamflow time series at a combination of different resolutions. Thus it is of interest to know what percentage of the record is comprised of subdaily data. We calculate the number of years of subdaily data for each station as all years that have at least 200 days of subdaily data. Figure 2 shows the distribution of the subdaily record percentage in our dataset. Around 100 stations have subdaily data percentages over 90 %. The other stations have percentages of subdaily data that range from 20 % to 90 %. Any station that has less than half of its record made up of subdaily data was manually validated to ensure that the sampling frequency adequately captured flood peaks at those locations. The stations showing a low percentage of subdaily data are characterized by having a long total record length compared to the subdaily record length, i.e. in these cases, the amount of subdaily data is not below average; rather, the overall record length is extensive. There was no correlation between model performance at the 1-hour duration and either total record length or percentage of the record that was subdaily data for each of the model evaluation metrics used in this study (results not shown).

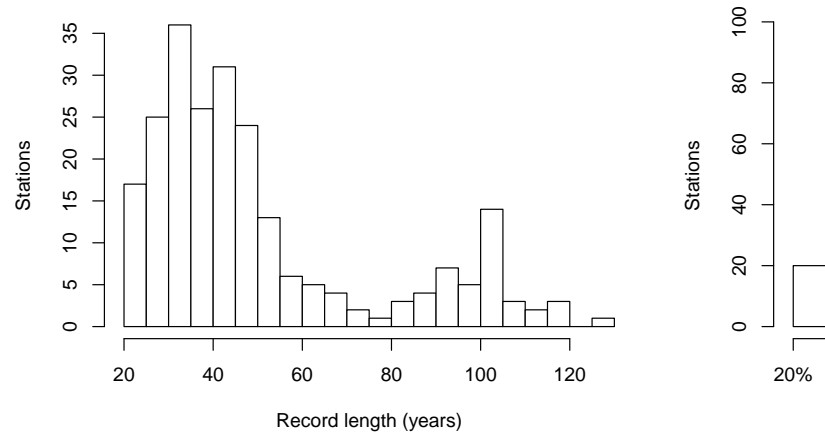 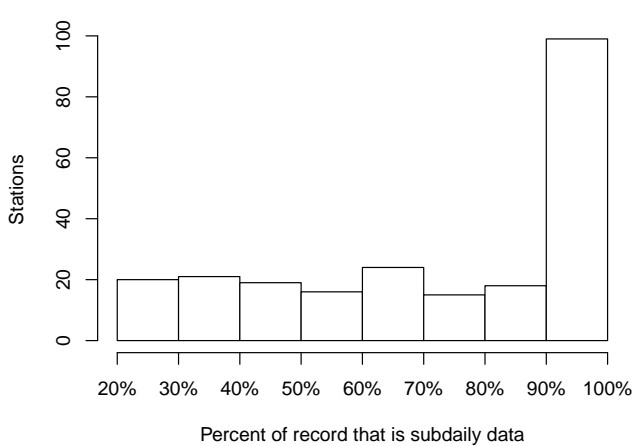

**Figure 2.** Histograms for record length (years) and percent of the record that is subdaily data. Only years that had at least 200 days of subdaily data count towards the subdaily data total when calculating the record percentage. Stations with less than 50 % of the record comprised of subdaily data were manually validated to make sure the sampling frequency of the data was high enough to represent flood peaks at that location.

In addition to quality control on the sampling frequency, the data have undergone a detailed quality control by the hydrometric section at NVE. Ice jams pose a challenge at numerous stations in Norway and can affect the accuracy of the rating curves used to estimate streamflows from water level measurements. In such cases, specific correction procedures outlined in NVE's internal quality assurance protocols have been implemented to obtain accurate discharge values. Any year with less than 300 days of data was excluded from the analysis.





## 2.2 Catchment descriptors

In addition to streamflow, we derived a set of geographical and hydro-climatic catchment descriptors for each catchment. Table 1 lists the 76 total catchment descriptors. The geographical descriptors include size and shape of the catchments, length of the river networks, land use properties and elevation distributions. The hydro-climatic descriptors include mean annual and monthly runoff, precipitation, temperature and sum of rain and snow melt. Note that all the hydro-climatic descriptors were based on interpolated observations given by the SeNorge 2.0 dataset (Lussana et al., 2019). Rain was defined as precipitation

when the temperature is above 0 °C. Snow melt was extracted from the SeNorge snow model (Saloranta, 2014).

For this study, we determined the average contribution of rainfall to floods at each catchment by calculating the ratio of rainfall to the total water depth, where the total water depth includes both rainfall and snowmelt accumulated within a specific time period prior to each flood. These ratios were then averaged across all flood events for the catchment. For details see Engeland et al. (2020).

The catchment areas vary from 0.52 km$^2$ to 6182 km$^2$, with a median size of 124 km$^2$. Roughly half (53 %) of the catchments have more than 1 % of their area covered by lakes; of these catchments, the median effective lake percentage is 2.8 %. Mean annual precipitation ranges from 390 mm to 3196 mm, displaying a notable east-west gradient across the country, with higher precipitation levels along the west coast. The mean annual temperature ranges from -4.0 °C to 7.2 °C, with a median of 0.15 °C. Temperature is influenced by both elevation and latitude; temperature decreases as elevation and latitude increase. The

minimum altitude of the catchments spans from sea level to 1104 m.a.s.l., while the catchment relief varies from 54 m to 2019 m. Catchments with greater relief are typically located in the mountain ranges along the west coast, which exhibits more rugged topography than the flatter regions of the country in the east.

## 3 Study design

A flowchart of the study design is displayed in Fig. 3. Panel (a) details the process of predictor selection for the GAM developed

in this study (*floodGAM*). The predictor selection process is divided into two parts. The first part (Part I, predictor pre-selection) is complementary to, but not necessary for, the second part. We detail our approach to predictor pre-selection in section 4.1, and note that other variable selection techniques or expert judgement could replace our approach detailed in this section. Part II, selection of predictors for floodGAM, is described in section 4.2. Panel (b) details the validation and visualization process for floodGAM. In the validation step, floodGAM is compared to the two benchmark models, RFFA_2018 (the existing

log-linear model for median flood estimation in Norway) and XGBoost. The benchmark models are summarized in section 4.4. We assess the performance of the models through a cross-validation study, such that predictive accuracy and reliability are assessed through the consistency between predictions and holdout data. Predictive performance for floodGAM and RFFA_2018 is assessed on five evaluation metrics (section 4.5). XGBoost provides a supplementary benchmark value for the mean absolute error (MAE); due to distributional assumptions, we cannot obtain optimal predictors for XGBoost for the other four evaluation

metrics. Reliability is assessed through the probability integral transform (PIT) which is also only available for RFFA_2018 and floodGAM. Predictive performance results are reported in section 5.1. Relibability results are reported in section 5.2. Finally,



**Table 1.** Descriptions of the 76 catchment descriptors used in the study, grouped into geographical and hydro-climatic descriptors. Abbreviations are further used in the text and figures.

| Variable | Description | Unit |
|---|---|---|
| $A$ | Logarithm of catchment area | km$^2$ |
| $O$ | Catchment circumference | m |
| $A_P$ | Catchment area / circumference * 1000 | km |
| $D, D_{net}$ | Drainage density (total river length / area), (total river length excluding lakes / area) | - |
| $C_L$ | Logarithm of catchment length | km |
| $R_L$ | Length of main river | km |
| $R_{TL}, R_L$ | Total river length, and total river length excluding lakes | km |
| $R_G, R_{G1085}$ | Gradient of main river, and gradient of main river excluding the 10 % lowest and 15 % highest reaches | m/km |
| $H_{10}, H_{50}, H_{90}$, | The 10th, 50th, and 90th percentile of the hypsographic curve, | |
| $H_{MAX}, H_{MIN}$ | maximum elevation, minimum elevation | m.a.s.l. |
| $H_F$ | Catchment relief (maximum elevation - minimum elevation) | m |
| $C_S$ | Mean slope | ° |
| $A_{Glac}, A_{Agr}, A_{Bog}, A_U$, | Percentage of catchment covered by glaciers, agriculture, bogs, urban areas, | % |
| $A_L, A_{LE}, A_{For}, A_{Mount}$ | lakes, effective lake percentage, forests, mountains | |
| $Q_N$ | Mean annual runoff 1961-1990 | l/s/km$^2$ |
| $P_{Jan}, P_{Feb}, P_{Mar}, P_{Apr}, P_{Mai}, P_{Jun}$, | Mean precipitation from 1961-1990 in January, February, March, April, May, June, | mm/month |
| $P_{Jul}, P_{Aug}, P_{Sep}, P_{Oct}, P_{Nov}, P_{Dec}$ | July, August, September, October, November, December | |
| $P_N$ | Mean annual precipitation 1961-1990 | mm/year |
| $P_{Med1Max}, P_{Med2Max}, P_{Med3Max}, P_{Med4Max}, P_{Med5Max}$ | Median of annual 1-, 2-, 3-, 4-, and 5-day precipitation | mm/day |
| $T_{Jan}, T_{Feb}, T_{Mar}, T_{Apr}, T_{Mai}, T_{Jun}$, | Mean temperature from 1961-1990 in January, February, March, April, May, June, | °C |
| $T_{Jul}, T_{Aug}, T_{Sep}, T_{Oct}, T_{Nov}, T_{Dec}$ | July, August, September, October, November, December | |
| $T_N$ | Mean annual temperature 1961-1990 | °C |
| $W_{Jan}, W_{Feb}, W_{Mar}, W_{Apr}, W_{Mai}, W_{Jun}$, | Mean sum of rainfall and snowmelt from 1961-1990 in January, February, March, April, May, June, | mm/month |
| $W_{Jul}, W_{Aug}, W_{Sep}, W_{Oct}, W_{Nov}, W_{Dec}$ | July, August, September, October, November, December | |
| $W_N$ | Mean annual sum of rainfall and snowmelt 1961-1990 | mm/year |
| $W_{Med1Max}, W_{Med2Max}, W_{Med3Max}, W_{Med4Max}, W_{Med5Max}$ | Median of annual 1-, 2-, 3-, 4-, and 5-day rainfall and snowmelt | mm/day |

section 5.3 presents the visualization and comparison of the data-driven relationships between predictors and the response identified by floodGAM across different durations.

# 4 Methods

## 4.1 Machine learning based pre-selection

The variable selection algorithm used is the Iterative Input Selection (IIS) algorithm proposed in Galelli and Castelletti (2013). The IIS algorithm selects a non-redundant set of variables using a ranking procedure and a stepwise forward selection process. Candidate variables are ranked using an input ranking algorithm, and the top-ranked variables are evaluated by adding them to the selected variable set and measuring prediction accuracy on a chosen model. This process is repeated with residuals as the new response variable until the best variable is already in the set or the model's performance does not improve. These two criteria—checking for repeated selection of variables or requiring improvement of predictive performance above a certain





**Figure 3.** Flowchart of the study design. Panel (a) details the process of predictor selection for the GAM developed in this study. Panel (b) details the model validation and visualization process.

threshold—constitute an automatic stopping condition for the algorithm. The algorithm in full can be found in Galelli and Castelletti (2013).



IIS requires the choice of (i) an input ranking algorithm and (ii) a model to evaluate the predictive performance of the
chosen subset of candidate variables. A tree-based ensemble is an effective choice for both (i) and (ii) since the ensemble
can be directly exploited as an input-ranking procedure; the particular structure of tree-based ensembles can be used to infer
the relative importance of input variables and order them accordingly. However, in the context of our analysis, choosing a
tree-based ensemble for (ii) assumes that relationships generated by the tree-based ensemble are simple enough to be able to
be modeled by the GAM. To take this into account, the tree depth—an important parameter controlling the interaction depth
between input variables—is set to one, since we do not consider variable interactions in the GAM. We choose XGBoost as the
tree-based ensemble within IIS in line with the recent work of Alsahaf et al. (2022). XGBoost is a popular open-source software
implementation of extreme gradient tree boosting (Chen et al., 2015; Chen and Guestrin, 2016), which was first proposed in
Friedman et al. (2000) and is a computationally efficient implementation of the gradient tree boosting from Friedman (2001).
Details of the algorithm set up and hyperparameter tuning for XGBoost can be found in Appendix C.

We run the IIS algorithm within a resampling method to assess consistency of the selected variable sets. This is important as
there is no uncertainty associated with the XGBoost output or the selected variable sets from IIS. For the resampling step in this
study, we choose to systematically resample without replacement, splitting our data into ten non-overlapping folds; however,
we note that other resampling methods, such as bootstrap, could also be used as the resampling step. This repeated application
of IIS to subsampled data means each application of the algorithm could potentially select a different variable set, where both
the chosen variables and the total number of variables are allowed to vary. A visual explanation of the IIS algorithm within
the resampling method can be found in Appendix C. The procedure is repeated once for the 1 hour duration and once for the
24 hour duration such that we can assess the consistence of selected variables across durations as well as across data folds. In
total, the IIS algorithm is applied to 10 subsampled data sets × 2 durations for a total of 20 applications. The consistency of
the selected variable sets is assessed across these 20 applications.

The IIS algorithm was run with all 76 catchment descriptors as input. Of those 76 variables, 34 were selected by at least one
of the folds and 7 were selected by at least five folds. The subset of variables that appeared most consistently, i.e. those that
were selected in at least five of the folds, is depicted in Fig. 4. The complete set of variables identified by IIS can be found in
Appendix C. In Fig. 4, the horizontal axis represents the number of times a variable was chosen within the resampling scheme,
out of a maximum possible of 10 (once for each fold). The color of the grid cells represents the order of variable selection
within the IIS algorithm. Variables selected first tend to be those that are most informative. For example, the predictor variable
$Q_N$ (mean annual runoff from 1961-1990 [l/s/km$^2$]) consistently emerges as the most important variable across all folds and
durations: every fold of subsampled data chooses $Q_N$ as the first predictor. Both consistency of inclusion and order of variable
selection can be considered when choosing which predictors to carry forward for more formal variable selection within the
model architecture of the GAM.

The catchment descriptors that were most consistently selected include $Q_N$, $A_{LE}$, and $A_P$. The catchment descriptor $A_L$,
which is highly correlated to $A_{LE}$, is also included for a minority of folds, but its inconsistent inclusion sets it apart from
$A_{LE}$. The other three less consistently selected predictors are $H_F$, which describes the difference in elevation from the highest
to lowest point [m]; $R_{G1085}$, which describes the gradient of main river excluding the 10 % lowest- and the 15 % highest





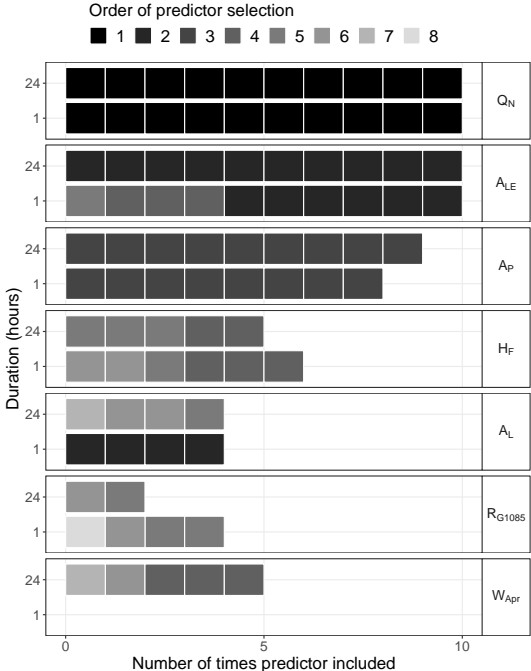

**Figure 4.** Variables from the pre-selection scheme that appear in at least 5 of the folds. The horizontal axis represents the number of times a variable was chosen. The vertical axis indicates the duration that generated the covariate set. The color indicates the order of variable selection within the IIS algorithm. Variables selected first tend to be those that are most informative.

reaches [m/km]; and $W_{Apr}$, which describes the mean sum of rain and snow melt in April from 1961-1990 [mm/month]. The two different durations generally selected the same predictors, particularly on those that were most consistently selected ($Q_N$, $A_{LE}$, and $A_P$). Duration specific differences beyond this should not be over interpreted given the variability in the full selected set shown in Appendix C.

### 4.2 Predictor selection for floodGAM

The predictors chosen for floodGAM are identified in Table 2 and include four geographical catchment descriptors: (1) $A_{LE}$ - effective lake percentage, (2) $A_P$ - the area of a catchment divided by its circumference, (3) $R_{G1085}$ - the gradient of the main river excluding the 10 % lowest and 15 % highest reaches, (4) $H_F$ - the difference in catchment elevation from the highest to lowest point, as well as three hydro-climatic catchment descriptors: (5) $Q_N$ - the mean annual runoff, (6) $W_{Apr}$ - the mean sum of rainfall and snowmelt in April, and (7) $P_{Sep}$ -the mean precipitation in September. These predictors were chosen using the results from the IIS algorithm (Fig. 4) in combination with expert judgement. The climate descriptor $P_{Sep}$ was not chosen by IIS but was added because the autumn- and winter flood season–with mainly rainfall-driven floods–is important in Norway and $P_{Sep}$ is a good representation of the autumn precipitation. The predictor $R_{G1085}$ is heavily skewed; we found it useful to



log transform that predictor for a more numerically stable estimation within floodGAM. All seven predictors were verified as significant by shrinkage estimation within the implementation of floodGAM.

One of the benchmark models, RFFA_2018, is the log-linear model currently used by NVE to predict the median flood
(Engeland et al., 2020). The RFFA_2018 model was developed for 24 hour flood data. Table 2 displays the catchment descriptors, and their transformations, used in RFFA_2018 and floodGAM. Predictors and predictor transforms in RFFA_2018 were chosen according to internal protocols at the Norwegian Water and Energy Directorate. The other benchmark model, XGBoost, has access to all 76 catchment descriptors. Previous research (Alsahaf et al., 2022) observed improved predictive performance with XGBoost models when employing IIS-based pre-selection; however, our analysis found that pre-selection applied to the
XGBoost models in this study did not alter the statistical significance of the results. For simplicity, all reported intervals and evaluation metrics pertain to the XGBoost model applied to the full catchment descriptor set.

**Table 2.** Descriptions of predictors used in the models floodGAM and RFFA_2018, structured into geographical (top) and hydro-climatic (bottom) catchment descriptors. Abbreviations are further used in figures. Inclusion of predictor variables is indicated for each model, and variable transformations are listed in their respective rows.

| Variable | Description | floodGAM | RFFA_2018 |
|---|---|---|---|
| $R_L$ | Length of main river [km] | | sqrt(x) |
| $A_{LE}$ | Effective lake percentages [ %] | x | x |
| $A_P$ | Catchment area / circumference * 1000 [km] | x | |
| $R_{G1085}$ | Gradient of main river excluding the 10 % lowest- and the 15 % highest reaches [m/km] | log(x) | |
| $H_F$ | Maximum elevation - minimum elevation [m] | x | |
| $Q_N$ | Mean annual runoff 1961-1990 [l/s/km$^2$] | x | $x^{1/3}$ |
| $T_{Feb}$ | Mean temperature February 1961-1990 [°C] | | $x^2$ |
| $T_{Mar}$ | Mean temperature March 1961-1990 [°C] | | $x^3$ |
| $W_{Mai}$ | Mean sum of rain and snow melt May 1961-1990 [mm/month] | | sqrt(x) |
| $W_{Apr}$ | Mean sum of rain and snow melt April 1961-1990 [mm/month] | x | |
| $P_{Sep}$ | Mean precipitation September 1961-1990 [mm/month] | x | |

## 4.3 Generalized Additive Models

GAMs, introduced by Hastie and Tibshirani (1987), are a class of regression models that extend the linear regression model to handle non-linear relationships between the predictor variables and the response variable. GAMs model the relationship
between the response variable and each predictor separately by assuming a smooth, continuous, non-parameteric function of each predictor. These functions are then combined additively to obtain the overall prediction. This allows for a wide range of predictor-response relationships to be captured without specifying a prior functional form. Furthermore, these predictor-response relationships are easily visualized by plotting the partial response curve for each predictor.





Let $\mathbf{y}$ be a vector of our response variable (the median flood at location $i$) with index $i \in [1,\ldots,n]$ referring to the $i$th element. Then the GAM relates the mean response for observation $i$ to the sum of smooth functions of $p$ explanatory variables $x_{i1},\ldots,x_{ip}$ as follows:

$$g\left(\mathbf{E}[y_i]\right) = \alpha + \sum_{j=1}^{p} s_j\left(x_{ij}\right) \tag{1}$$

where $s_j()$ is the smooth function of predictor $x_{ij}$, $\alpha$ is the intercept and $g()$ is a monotonically differentiable link function. The smooth function $s_j()$ is defined by a linear combination of basis functions, allowing the relationship between $x_{ij}$ and response $y_i$ to be non-parametrically modeled. Because this non-parametric construction is so flexible, selecting the appropriate level of 'smoothness' for each predictor is an important component of GAM construction. In practice, this is often done by limiting the effective degrees of freedom. We used a thin plate spline basis with effective degrees of freedom limited between 6 and 3 for our chosen predictors.

While the form of the predictor-response relationship can be modeled non-parametrically, the probability distribution of the response variable in the GAM must still be specified. We chose to model the data as normally distributed with a log link, in line with standard practices in hydrology that model flood volumes and flood peak discharges as log normal (Stedinger, 1980).

We used the 'mgcv' package in the R statistical software (Wood, 2017) to implement the GAMs. The mgcv package contains a convenient variable selection method based on null-space penalization, which allows smooth functions associated with a particular predictor to be penalized to the zero function and thereby selected out of the model if the predictor is nonimportant (Marra and Wood, 2011). This capability is accessed by setting the *select* argument of the gam() function to 'True'. We set select = T and use restricted maximum likelihood ('REML') as the estimation method for each of the GAMs in this study.

### 4.4 Benchmark models

#### 4.4.1 RFFA_2018

The existing model for index flood estimation in Norway (RFFA_2018) is the log-linear model presented in Engeland et al. (2020). Let $\mathbf{y}$ be a vector of our response variable with index $i \in [1,\ldots,n]$ referring to the $i$th element. Let $\mathbf{X}$ be our predictor matrix with $n \times p$ elements, where $p$ is the number of predictor variables. Furthermore, since we wish to evaluate the predictor-response relationship on the log scale, let $z_i = \log(y_i)$. Then the regression equation is given as

$$z_i = \alpha + \sum_{j=1}^{p} \beta_j x_{ij} + \epsilon, \tag{2}$$

where $\alpha$ and $\beta_j$ are the parameters to be estimated and $\epsilon$ is the error term that is assumed normally distributed $N(0,\sigma^2)$. The mgcv package provides routines to fit log-linear models as well as GAMs and was used to estimate RFFA_2018 in this study.





### 4.4.2 XGBoost

XGBoost is a popular open-source software implementation of extreme gradient tree boosting (Chen et al., 2015; Chen and Guestrin, 2016), which was first proposed in Friedman et al. (2000) and is a computationally efficient implementation of the gradient tree boosting from Friedman (2001).

Gradient tree boosting is a machine learning technique that involves training an ensemble of decision trees sequentially, with each subsequent tree aimed at reducing the residual errors of the previous tree. At each step, a gradient descent algorithm is used to optimize a predefined loss function by adjusting the weights of the features in each tree.

     Let $\mathbf{y}$ be a vector of our response variable with index $i \in [1, \ldots, n]$ referring to the $i$th element. Let $\mathbf{X}$ be our predictor matrix with $n \times p$ elements, where $p$ is the number of predictor variables. Furthermore, since we wish to evaluate the predictor-response

relationship on the log scale, let $z_i = \log(y_i)$. Then the regression equation is given as

$$\hat{z}_i = \sum_{k=1}^{K} f_k(\mathbf{x}_i), \; f_k \in \mathcal{F}, \tag{3}$$

where $f_k$, $k \in [1, \ldots, K]$ is the ensemble of regression trees and $K$ is the number of trees used. Here $\mathbf{x}_i$ is the $i$th row of the predictor matrix and $\mathcal{F}$ is the set of all possible classification and regression trees (e.g. CARTs; see XGBoost documentation). Then the objective function to be minimized is given by

$$L^k = \sum_{i=1}^{n} L\left(z_i, \hat{z}_i^{k-1} + \eta f_k(\mathbf{x}_i)\right) + \Omega(f_k) \tag{4}$$

where $L^k$ is the $k$th iteration loss, $\hat{z}_i^{k-1}$ is the prediction at the previous iteration, $\eta$ is a shrinkage parameter controlling the learning rate, $f_k$ is the tree that provides the best improvement to the model as measured by the predefined loss function, and $\Omega(f_k)$ is a penalization parameter that controls the complexity of trees to avoid overfitting. Here we used the squared error loss as the objective function:

$$L = \sum_{i=1}^{n} (z_i - \hat{z}_i)^2. \tag{5}$$

     The following hyperparameters were tuned on the indicated ranges: tree depth (1-10); the percentage of observations subsampled at each boosting step (0.1-1); the minimum number of instances needed in each node (1-5); and the shrinkage parameter $\eta$ (0.01-0.1). The number of boosting iterations was evaluated up to a maximum number of 999 iterations. Hyperparameter tuning was conducted within a 10-fold cross-validation scheme using all possible parameter combinations and an early stop-

ping criterion for the number of boosting iterations, where the algorithm stopped after 25 rounds without improvement in the error rate. The ranges of the hyperparameters were chosen based on experience with the data set and recommended XGBoost practices.

### 4.5    Evaluation methods

This section presents (i) the error metrics used to evaluate the predictive performance of the models, (ii) a computationally

efficient permutation test that allows us to assess the statistical significance of differences in error metrics between the models



(Thorarinsdottir et al., 2020) and (iii) the probability integral transform (PIT). The PIT is used to assess reliability of the models as measured by the consistency between model predictions and validation data. We assess the performance of the models through a cross-validation study, such that predictive accuracy and reliability are assessed through the consistency between predictions and holdout data.

### 4.5.1 Error metrics

We evaluate model performance using the root mean squared error (RMSE), the mean absolute error (MAE), the mean relative error (MRE), the mean absolute percent error (MAPE) and the continuous ranked probability score (CRPS) (Gneiting and Raftery, 2007; Hersbach, 2000). All of these metrics measure slightly different aspects of the predictive distribution. The RMSE, MAE, and CRPS are expressed in the units of the response variable (l/s/km2) and give more weight to catchments with higher discharge values. In our case, these metrics tend to prioritize minimizing errors in catchments located on the west coast of Norway, where the median flood values, given in [l/s/km$^2$], are the highest. The proportional error metrics–the MAPE and the MRE–avoid this issue of scale but are sensitive to highly over- or under-estimated values. Four of the metrics here (RMSE, MAE, MRE, MAPE) assess the distance between an observed value and a single predicted value; that is, they are error metrics for point forecasts. The CRPS measures the difference between the predicted and observed cumulative distributions (Hersbach, 2000) and thus provides a measure of how variable the predictions are in addition to assessing accuracy.

Constructing a statistically meaningful model ranking from these error metrics requires that the predicted value from the model minimizes the given error metric. For example, the root mean squared error (RMSE) is minimized when the predicted value is chosen as the mean of the predictive distribution. If an alternative distributional feature, such as the median, is used with the RMSE in the situation where the median and mean of the predictive distribution are not equivalent (e.g., when the data are assumed log normal), any model rankings constructed from the resulting quantity will be unreliable.

We list the optimal predictor (minimizing quantity) for each of the error metrics for point forecasts in Table 3. The MRE and MAPE are minimized by the functionals given in Gneiting (2011); see Appendix A for calculation of the optimal predictors here. Both floodGAM and RFFA_2018 are assessed on all five metrics. XGBoost is assessed only on the MAE; optimal predictors for the other four error metrics are not accessible for XGBoost when the data are assumed log normal. Table 3 defines the metrics and reports the associated units. All metrics are negatively oriented, i.e. a smaller value indicates better predictive performance.

### 4.5.2 Permutation test

The permutation test, as defined in Thorarinsdottir et al. (2020), determines the difference in scores between two models, $M_1$ and $M_2$, by computing

$$c = \frac{1}{n} \sum_{i=1}^{n} \left( \phi(M_1) - \phi(M_2) \right) \tag{6}$$

Here, $n$ represents the total number of stations and $\phi(\cdot)$ is a scoring measure (for example, the absolute error or the percent absolute error). If $c$ is negative, it indicates that model $M_1$ performs better than model $M_2$ in terms of the scoring measure,



**Table 3.** Definitions of error metrics used in this study: root mean squared error, mean absolute error, mean relative error, mean absolute percent error, and continuous ranked probability score. Here $\hat{y}_i$ is the predicted value at station $i$, $i \in [1, \ldots, n]$, $y_i$ is the observed value at station $i$, and $F$ is the cumulative distribution function of the predictive distribution with finite first moment. For the CRPS, $H(\hat{y}_i - y_i)$ denotes the Heaviside function and takes the value 0 when $\hat{y}_i < y_i$ and the value 1 otherwise.

| Error metric | | Optimal predictor | Units |
|---|---|---|---|
| RMSE | $\sqrt{\left(\frac{1}{n}\sum_{i=1}^{n}(y_i - \hat{y}_i)^2\right)}$ | $\hat{y}_i = \text{mean}(F)$ | l/s/km2 |
| MAE | $\frac{1}{n}\sum_{i=1}^{n}|y_i - \hat{y}_i|$ | $\hat{y}_i = \text{median}(F)$ | l/s/km2 |
| MAPE | $\frac{1}{n}\sum_{i=1}^{n}\left|\frac{y_i - \hat{y}_i}{y_i}\right| \cdot 100$ | $\hat{y}_i = \text{med}^{(-1)}(F)$ | % |
| MRE | $\frac{1}{n}\sum_{i=1}^{n}\left|\frac{y_i - \hat{y}_i}{\hat{y}_i}\right| \cdot 100$ | $\hat{y}_i = \text{med}^{(1)}(F)$ | % |
| CRPS | $\int_{-\infty}^{\infty}\left[F(\hat{y}_i) - H(\hat{y}_i - y_i)\right]^2 dy_i$ | - | l/s/km2 |

and vice versa. The permutation test creates resampled copies of $c$ with randomly swapped models $M_1$ and $M_2$. Under the null hypothesis that both models perform equally well, the set of permutations cannot be differentiated from $c$. The statistical test

formalizes this concept by determining which quantile $c$ occupies in the set of permutations; if the p-value is less than 0.05, then it suggests that the performance of $M_1$ is significantly better than $M_2$.

### 4.5.3 Probability integral transform

Reliability describes the consistency between model predictions and validation data. A reliable model is expected to produce an estimated distribution that closely aligns with the unknown true distribution of the data. This is typically assessed through the

probability integral transform (PIT); if the observations follow the estimated distribution, the PIT values will be approximately uniformly distributed (Dawid, 1984):

$$\hat{F}(y_i) \;\overset{\cdot}{\sim}\; U([0,1]).$$

The uniformity of the PIT values is represented graphically through histograms. As the PIT requires the estimated cumulative distribution of the predictive distribution, $\hat{F}$, the reliability assessment is not accessible for the XGBoost models in this study.

## 5   Results

### 5.1   Predictive performance

We report the performance evaluation metrics for floodGAM, RFFA_2018, and XGBoost in Table 4. The best result is shown in bold font. If floodGAM was statistically significantly better than RFFA_2018 at the $\alpha = 0.05$ level on a particular metric and duration, the significance is indicated with an asterisk. The predictive performance for floodGAM predicting across durations–



that is, using floodGAM fit on the 24 hour data to predict at the 1 hour duration and vice versa–is shown and the duration used
to fit the model is indicated in the model name ("floodGAM, 24 hours" and "floodGAM, 1 hour").

On the 1 hour duration, floodGAM was statistically significantly better than RFFA_2018. There were no statistically significant differences between floodGAM and RFFA_2018 on the 24 hour duration, or between floodGAM and XGBoost on either the 1 hour or 24 hour MAE. The floodGAM predicting across durations was not competitive and was statistically significantly
worse than the duration-specific floodGAM.

**Table 4.** Model evaluation metrics–root mean squared error, (mean) continuous ranked probability score, mean absolute error, mean relative error, mean absolute percent error–showing predictive performance for floodGAM and the benchmark models. The best result is shown in bold font. If floodGAM was statistically significantly better at the $\alpha = 0.05$ level than RFFA_2018 on a particular metric and duration, the significance is indicated with an asterisk.

| Duration | Model type | Name | Evaluation metric | | | | |
|---|---|---|---|---|---|---|---|
| | | | RMSE [l/s/km2] | CRPS [l/s/km2] | MAE [l/s/km2] | MRE [%] | MAPE [%] |
| 1 hour | GAM | floodGAM | **122.2\*** | **61.2\*** | **84.9\*** | **20.4\*** | **20.5\*** |
| | | floodGAM, 24 hours | 172.7 | 76.0 | 104.4 | 23.4 | 25.9 |
| | Log-linear | RFFA_2018 | 137.7 | 69.5 | 97.0 | 25.3 | 24.2 |
| | ML | XGBoost | - | - | 89.9 | - | - |
| 24 hours | GAM | floodGAM | **84.1** | **42.7** | **59.5** | **17.0** | **17.5** |
| | | floodGAM, 1 hour | 157.3 | 72.2 | 102.3 | 22.8 | 20.4 |
| | Log-linear | RFFA_2018 | 85.9 | 43.8 | 61.8 | 17.4 | 17.6 |
| | ML | XGBoost | - | - | 60.5 | - | - |

To illustrate how floodGAM improves on RFFA_2018 at the 1 hour duration, we plot a model-by-model comparison of the error at each station (Fig. 5). Here we show three different error metrics (absolute percent error, relative error and the CRPS). Figures for other metrics, models and durations can be found in Appendix D. Points falling above the diagonal line indicate stations where RFFA_2018 performed worse than floodGAM. Points falling below the diagonal line indicate stations
where floodGAM performed worse than RFFA_2018. Point size shows catchment area, point color indicates the fraction of rain contribution to flood.

Figure 5 shows that RFFA_2018 systematically underestimates the 1 hour median flood in large, snowmelt driven catchments (Panel (a) of Fig. 5). The fact that these large, snowmelt driven catchments have relative errors that are greater than the absolute percent error means that the observed values are higher than the predicted values: RFFA_2018 is underestimating at these
stations. This underestimation is not obvious when looking at the absolute percent error as the absolute percent error supports severe underestimation (Gneiting, 2011). The opposite effect–supporting severe overestimation–is true for the relative error. In addition to improved performance in extreme cases, Fig. 5 shows that floodGAM has better performance in the bulk of the





data; that is, there is a higher density of points above the diagonal. This is visualized through the kernel density estimation underlaid on each panel (shaded areas in Fig. 5). Panel (b) shows the catchments with high CRPS values are typically rain-
driven catchments with small area and large discharge values.

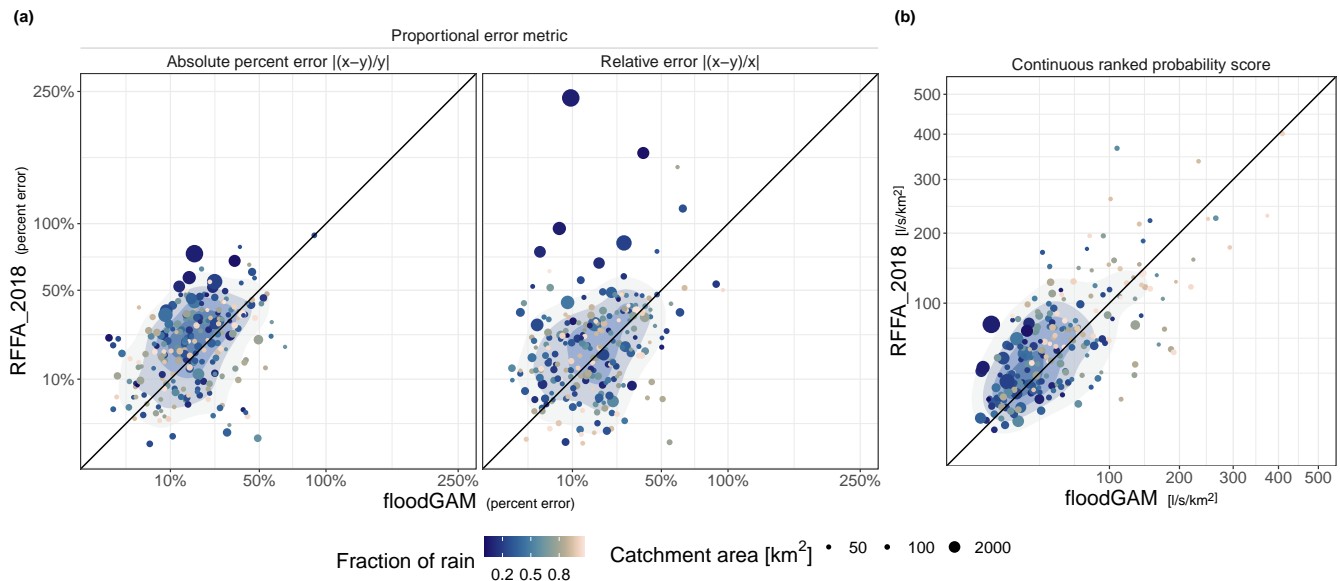

**Figure 5.** Model to model comparison on absolute percent error, relative error, and the continuous ranked probability score for RFFA_2018 and floodGAM on the 1 hour duration. In the panel headers, $x$ represents the predicted value and $y$ the observed value. Points falling above the diagonal line indicate stations where RFFA_2018 performed worse than floodGAM. Points falling below the diagonal line indicate stations where floodGAM performed worse than RFFA_2018. The 2D kernel density estimation of point density is underlaid to aid visual interpretation. Point size shows catchment area, point color indicates the fraction of rain contribution to flood.

## 5.2   Model reliability

We assess the reliability of the predictions for floodGAM and the existing model, RFFA_2018. Figure 6 shows histograms for floodGAM and RFFA_2018 at both the 1 and 24 hour durations. Histograms for both models are roughly uniform but show some evidence of bias: RFFA_2018 has an excess of values at high quantiles, while floodGAM has an excess of values
at low quantiles. The bias in RFFA_2018 shows a tendency to underestimate predicted values; this is consistent with results shown by the model evaluation metrics in Section 5.1, where the largest relative errors were caused by underestimations at large, snowmelt driven catchments. From the PIT histogram we see that the bias in floodGAM, on the other hand, tends toward overestimation, although the evaluation metric assessment in Fig. 5 shows none of the overestimations were as severe as the underestimations provided by RFFA_2018.
Table 5 shows the empirical coverage of the associated central 50 %, 80 % and 90 % prediction intervals. The empirical coverage is given by the area under the relevant number of central bins in the histograms in Fig. 6; for example, the 50 %





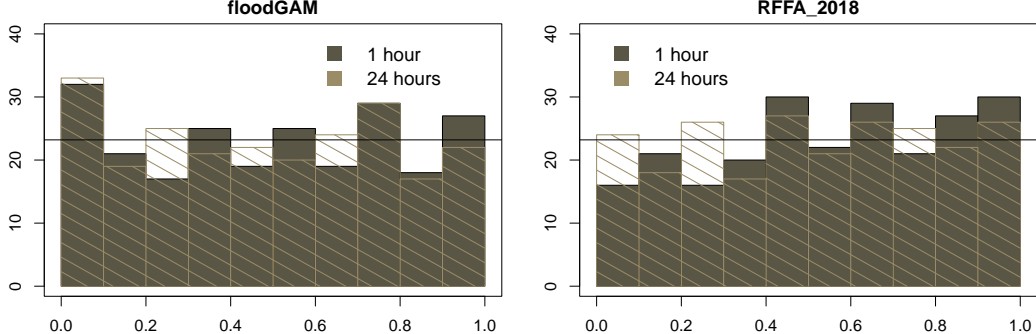

**Figure 6.** Visualizations of probability integral transform (PIT) values for floodGAM and RFFA_2018 at both the 1 and 24 hour durations. If the unknown true distribution of the data is close to the estimated distribution from the models, the PIT values will be uniformly distributed. This is assessed visually via histograms. Both distributions are roughly uniform but show some evidence of bias: RFFA_2018 shows an excess of values at the high quantiles, meaning values tend to be underestimated by this model. The floodGAM model has an excess of values at the low quantiles, indicating a tendency to overestimate.

empirical coverage is the total area under the central five bins in the PIT histogram. We replicate that information in numerical form in Table 5 for easy model to model comparison. The average width of the empirical prediction intervals, in l/s/km2, is also shown in Table 5. Wider prediction intervals indicate predictions that are less precise and therefore less informative.

Both models on both durations show empirical coverage probabilities that match the nominal coverage probabilities, reflecting the uniformity we see in the histograms in Fig. 6. However, we see differences in the average width of the probability intervals for the 1 hour duration. For this duration, the RFFA_2018 model reports intervals that are about 30 % wider on average than those for the floodGAM model. This shows the 1 hour predictions for RFFA_2018 are much less precise than those from floodGAM. The average width of the intervals between the two models is much more similar for the 24 hour duration, 450   although floodGAM still has narrower prediction intervals on average.

**Table 5.** Empirical coverage and average widths (in l/s/km2) of central prediction intervals for both floodGAM and RFFA_2018. The nominal coverage is 50 %, 80 % and 90 %. Within each duration, models are ordered according to their average CRPS scores (reported in Table 4).

| Duration | Model | 50 % | | 80 % | | 90 % | |
|---|---|---|---|---|---|---|---|
| | | Coverage | Width | Coverage | Width | Coverage | Width |
| 1 hour | floodGAM | 48.7 % | 143 | 74.6 % | 275 | 85.8 % | 356 |
| | RFFA_2018 | 52.2 % | 189 | 80.1 % | 367 | 92.7 % | 479 |
| 24 hours | floodGAM | 50.9 % | 103 | 76.2 % | 198 | 87.5 % | 256 |
| | RFFA_2018 | 50.9 % | 109 | 78.5 % | 208 | 90.1 % | 270 |



## 5.3 Explaining the model

The partial response curves for each predictor in floodGAM are plotted in Fig. 7. The partial response curves are the smooth components of floodGAM. They show how the median flood varies as a function of a particular predictor when all other predictors are held constant at their mean value. Note that the partial response curves in Fig. 7 are displayed on the link (log) scale, not in the units of the response. Therefore, predictor-response interpretation focuses on whether a predictor has an increasing or decreasing effect on the median flood, and the magnitude of this effect is assessed relative to other predictors in the model. The y axis range of the partial response curves in Fig. 7 indicates the relative importance of the predictors; a predictors that are more important have a larger range. Additional information about the relative importance of predictors can be obtained from formal measures such as the likelihood ratios between the full model and the model withholding particular predictors and is displayed in Table 6. Finally, the shading around the partial response curves is the estimation uncertainty associated with each smooth component. Areas with little data–for example, catchments with area to circumference ratios above 5 km–have large estimation uncertainty and possible forms of the smooth component can vary within this uncertainty interval.

Figure 7 also displays the partial residuals associated with each smooth component. Partial residuals for smooth components are the residuals that would be obtained by excluding the specific term from the model while keeping all other estimates fixed. The partial residuals used here are the working residuals from the 24 hour duration added to the corresponding estimate of the smooth term. Coloring the partial residuals by fraction of rain and sizing by catchment area can give a better idea of what types of catchments contribute to the shape of the smooth component. This can aid in identification of predictor-response relationships that are mechanistically realistic.

Figure 7 shows that the smooth component for $Q_N$ shows an increasing relationship whereas $A_{LE}$, $A_P$ and $P_{Sep}$ show a decreasing relationship with the median flood. Three smooth components $H_F$, $R_{G1085}$ and $W_{Apr}$ have a concave relationship with the median flood.

We see significant differences between durations in the smooth components for $A_{LE}$ and $A_P$; that is, these smooth components display segments where there is no overlap in the estimation uncertainty intervals between the 1 hour duration and the 24 hour duration. These two predictors that show duration-specific differences are important predictors. Table 6 reports the predictor ranking by likelihood ratio: $A_{LE}$ ranks as the most important predictor and $A_P$ the third most important predictor. This can also be assessed visually through the y-axis range of the smooth components in Fig. 7 (top row). We do not see significant differences between durations in the smooth component for the second most important predictor, $Q_N$. Additionally, the remaining four predictors do not show significant duration-specific differences. The least important predictor on both the 1 hour duration and 24 hour duration is $W_{Apr}$. The other climate variable–$P_{Sep}$–also has a low importance ranking and is ranked as second to last on the 24 hour duration and third to last on the 1 hour duration. Given the similarity of values between the lowest ranked predictors we do not see the reordering of $P_{Sep}$ between durations as significant.

When a predictor is correlated with covariates that describe spatial variation of the median flood in Norway, the partial residuals for that predictor show regional groupings (see, for example, $Q_N$, where the upper end of the smooth effect is







**Figure 7.** Partial response curves and partial residuals for floodGAM modeling the median annual maxima flood. The smooth components and partial residuals are shown on the link scale, and units for predictors are shown in panel titles. Partial residuals are colored by average fraction of rain contribution and sized by area. 95 % intervals showing estimation uncertainty for the smooth components are shaded. Location of data points for each predictor are shown as tick marks on the x axis. Y axis ranges span the same magnitude for each panel.

dominated by rainfall-driven catchments on the coast of Norway). However, we found no predictive performance benefit when splitting the data into hydrologically homogeneous regions, indicating that the GAM is flexible enough to adjust to differences in response effect between regions within Norway. Evaluation metrics, tests of statistical significance, and regional delineation for the assessment of the GAM on sub-regions are reported in Appendix E.





**Table 6.** Predictor ranking by likelihood ratio; larger values indicate greater predictor importance. Column "Absolute gain in likelihood value" is given by $-\log\left(L_0/L_{full}\right)$, i.e. the positive value of the difference in log likelihood between $L_0$, the model without a particular predictor, and $L_{full}$, the model with all seven predictors included. The ranking of predictors is consistent between the 1 and 24 hour durations, with the exception of $H_F$ and $P_{Sep}$.

| Predictor | Absolute gain in likelihood value | |
|---|---|---|
| | 1 hour | 24 hours |
| $A_{LE}$ | 129.6 | 87.7 |
| $Q_N$ | 83.7 | 85.9 |
| $A_P$ | 44.2 | 24.2 |
| $R_{G1085}$ | 22.6 | 15.3 |
| $H_F$ | 12.2 | 12.2 |
| $P_{Sep}$ | 12.6 | 7.7 |
| $W_{Apr}$ | 8.6 | 3.1 |

# 6 Discussion

We have, in accordance with our main objective, developed a GAM (floodGAM) such that we could identify and describe the functional relationships between the median flood and catchment descriptors at two different durations. Adequacy of floodGAM as an explainable model was established through predictive performance at ungauged locations, where predictive performance was measured by both predictive accuracy and reliability. The predictive accuracy and reliability of floodGAM matched or exceeded that of the benchmark models at both durations studied.

## 6.1 Hydrologic interpretation of predictor-response relationships in floodGAM

The shape of the smooth components should be interpreted with care: as with all statistical models, there is potential that the relationship reflects unidentified latent or confounding variables rather than a mechanistic relationship with the response. However, we can say the top three most important predictors ($Q_N$, $A_{LE}$, and $A_P$) have smooth components that are consistent with our expectations for the relationship with the median flood. The smooth component for $Q_N$ shows an increasing relationship;
as the mean annual runoff for a catchment gets larger so does the median flood. The smooth component for $A_{LE}$ shows a decreasing relationship with the median flood, reflecting the dampening effect of effective lake percentage on flood peak.

The smooth component for $A_P$ shows a decreasing relationship with the median flood. $A_P$ reflects both catchment size and shape. For catchment with similar shapes, $A_P$ increases with catchment area. For catchments with similar areas, $A_P$ is the largest for perfectly circular shapes and the smallest for elongated or irregularly shaped catchments. The circumference used
to calculate $A_P$ depends both on the approach used to calculate the catchment boundaries and the resolution of the underlying



digital elevation model. In our dataset we assume that the catchment boundaries are consistently defined as they all have a unique source (GeoNorge, 2021).

For our dataset, the catchment area explains most of the variation in $A_P$. The decreasing relationship between $A_P$ and median flood can therefore be explained by the well-known spatial scaling of floods (Alexander, 1972; Blöschl and Sivapalan, 1995;

Robinson and Sivapalan, 1997a, b; Tsonis et al., 2007; Tarasova et al., 2018; Stein et al., 2021; Najibi and Devineni, 2023). This scaling reflects the changing influence of runoff-generating processes based on catchment size (Blöschl and Sivapalan, 1995; Tarasova et al., 2018), as summarized in Lun et al. (2021): Firstly, a small catchment is more likely to be fully covered by a storm than a large catchment. Consequently, the variance of extreme catchment-average precipitation and thereby the median flood decreases with catchment size (Viglione et al., 2010). Secondly, there is a transition from short-duration convective events

to long-duration stratiform precipitation events as the most relevant flood generating process as catchment size increase (Gaál et al., 2015; Merz and Blöschl, 2009). In our data we see also that the snow melt contribution to floods increases with catchment size. Thirdly, the response times increase with area (Gaál et al., 2012) causing smaller flood peaks.

Relationships between catchment shape and flood size is less clear (Stein et al., 2021) and depends on how the time space organization of storm events interacts with the spatial organization of the catchments (Zoccatelli et al., 2011). Based on runoff

generation processes, Blöschl (2013) and Viglione and Blöschl (2009) argue that round catchments can be expected to react more quickly than elongated catchments since the flood waves from different parts of the catchment will concentrate quickly. On the other hand, a storm cell that follows a elongated catchments from the top towards the outlet might result in a high flood peak since the flood wave from upstream and downstream parts will overlap (Murthy, 2002). In an empirical study by David and Davidova (2014) the connections between catchment shape and flood magnitude are not significant.

In this study $A_P$ was consistently preferred as a predictor instead of other descriptors reflecting catchment size ($A$, $C_L$, $R_L$, $R_{TL}$, $R_{TL,net}$), indicating that the catchment shape influences the flood sizes. However, the marginal effect of catchment shape cannot be detected from our model.

Concave relationships between the smooth components $H_F$, $R_{G1085}$ and $W_{Apr}$ and the median flood are challenging to explain and might be a result of inter-correlated predictors and hidden variables. The smooth component for $P_{Sep}$ shows an

increasing linear relationship with the median flood. This is a reasonable relationship for the rainfall-driven catchments that experience high flows during autumn and winter; the partial residuals in Fig. 7 show the increasing nature of the smooth component is driven by catchments with a higher fraction of rain contribution to flood generating process. However, it is less clear that this increasing linear relationship should hold for the snowmelt-driven catchments that experience high flows during spring and summer.

This study was limited to constructing a model for annual maxima since flood guidelines in Norway pertain to annual maximum values; however, as a preliminary investigation into how seasonal flood regimes may influence the shape of the partial response curves shown in Fig. 7, we investigated changes in the partial response curves of floodGAM when seasonal maxima were used instead of annual maxima. Results are reported in Appendix F. We observed season-specific changes in the shape of the partial response curves for climatic variables. These changes were not observed in the partial response curves

for the geographical catchment descriptors or the mean annual runoff. This suggests relationships between climatic predictors





and annual maxima should be interpreted with caution as these relationships may represent a compromise between different generating processes. This parallels the observations in, for example, Ouarda et al. (2006), McCuen and Beighley (2003), and Fischer and Schumann (2021). Focusing on the role of climatic variables in regression style models that explicitly account for flood generating processes is an interesting area of future research for descriptive statistical studies, particularly when

investigating models that incorporate non-stationarities in climate: extrapolating any regression style model to future climates is problematic if the relationship between predictor and response is represented in a physically unrealistic way.

### 6.2    Duration-specific differences in median flood estimation

We observe duration-specific differences in the partial response curves for the predictors $A_{LE}$ (effective lake percentage) and $A_P$ (catchment shape). These differences can be described as changes in the predictors' magnitude of effect; that is, the y axis

range of the partial response curves for $A_{LE}$ and $A_P$ is larger for the 1 hour duration than the 24 hour duration. This means that floodGAM finds the influence of effective lake percentage and catchment shape on the median flood to be more pronounced at shorter durations.

     The results from the data-driven model (floodGAM) indicate that the relationship between catchment descriptors and the median flood changes with duration. This means that in order to optimally model each duration, the form of the functional

relationship between catchment descriptors and the median flood should be adapted and re-estimated at each duration.

     We examined how performance changes when these requirements are relaxed in various ways. First, to assess the performance when assuming a fixed relationship between median flood and predictors, we employed the floodGAM fitted on one duration to make predictions for another duration. We found that using the relationships established by floodGAM for one duration to predict for another led to diminished predictive performance. Secondly, to assess performance when assuming a

parametric relationship and re-estimating the model for each duration, we fit RFFA_2018—which was developed for the 24 hour data–to the 1 hour data. Once again, the performance was lower compared to the fully flexible duration-specific model, although assuming a fixed parametric form and re-estimating yielded better results than assuming an entirely fixed relationship (without re-estimating the coefficients). In the context of models that simultaneously estimate the median flood at different durations (e.g. regional QDF models), this suggests it would be challenging to achieve optimal outcomes for every duration. In

such scenarios, practitioners might need to decide on which durations reduced performance would be acceptable.

### 6.3    Predictor selection

Our study was focused on the question: does a data-driven model (floodGAM) detect duration-specific differences in how catchment covariates influence the median flood? If we gauge model adequacy through predictive performance, we are naturally confined to answering our question within predictor sets that work well with these sorts of data-driven models.

Identification of predictor sets that are good for data-driven models can be interesting in and of itself as it is possible that data-driven models can uncover predictor information that was previously unclear (Guyon and Elisseeff, 2003). The challenge here is that using a data-driven model for selection implies in most cases a model-based preselection, which is not guaranteed to generate a predictor set that will work within other model architectures (Maier et al., 2010). In this study, one type of data-





driven model (a boosted tree ensemble with a depth of one) was used to preselect a predictor set that was then validated inside
a different type of data-driven model architecture (the GAM). The selection of the predictor set by two different data-driven
models suggests some sort of robustness. However, we do not necessarily expect the chosen predictor set in this study to give
good results when used with an entirely different model architecture, e.g. a log-linear model.

This limits cross-model architecture and cross-predictor set questions. For example, we cannot say if the differences in
performance between floodGAM and RFFA_2018 at the 1 hour duration are due to the fixed functional form assumed in
RFFA_2018 or the differences in predictor sets, although the duration-specific differences identified within floodGAM suggest
that it is advantageous to be able to adapt to different predictor-response relationships at different durations. Answering ques-
tions focusing on the duration dependence of particular catchment descriptors or predictor sets is an interesting area of future
research that requires hydrology-specific knowledge reflecting a mechanistic understanding of the process at hand.

## 7 Conclusions

We develop a generalized additive modeling approach for estimation of the median annual maximum (index) flood, with a
focus on detection and description of the functional relationships between the median flood and catchment descriptors at
multiple durations. We employ a machine learning-based variable pre-selection tool to aid in predictor selection and increase
the practicality of constructing generalized additive models (GAMs) for index flood estimation. We establish the adequacy
of the GAM as an explainable model through predictive performance at ungauged locations, where predictive performance
was measured by both predictive accuracy and reliability. The predictive performance of the GAM developed in this study
(floodGAM) is compared to two benchmark models, the existing log-linear model for median flood estimation in Norway and
a fully data-driven machine learning model (an extreme gradient boosting tree ensemble, XGBoost). We find that

- The predictive accuracy and reliability of floodGAM matches or exceeds that of the benchmark models at both durations
  studied.

- We observe duration-specific differences in the form of the functional relationship between the median flood and two
  catchment descriptors (effective lake percentage and catchment shape) within the predictor set considered in floodGAM.
  Ignoring these differences results in a statistically significant decline in predictive performance.

If index flood estimation at multiple durations is the goal, these results suggest that it may be difficult to obtain optimal per-
formance on all durations when assuming a fixed or parametric form between predictors and response. Models and approaches
that make these assumptions while accounting for, or extrapolating to, different durations should consider on which durations
it would be acceptable to have reduced performance. Finally, in situations where predictive performance at multiple observed
durations is a priority, floodGAM emerges as a promising option. The ability to auto-adapt functional relationships at multi-
ple durations offers a potential simplification of the modeling process and could be a practical alternative to development of
separate parametric forms. Furthermore, the comparative predictive performance between floodGAM and XGBoost suggests





605  that floodGAM is adequately capturing the available relationships in the data, while also providing accessible information on
prediction uncertainty.

## Appendix A:  Computation of minimizing quantity for relative error and absolute percent error

The optimal predictor for the relative error is the functional $\mathrm{med}^{(1)}(F)$, defined in Gneiting (2011), which is the median of the
distribution with density proportional to $x f(x)$. Here $f(x)$ is the probability density function for the log normal distribution;

610  that is:

$$
f(x) = \frac{1}{x \sigma \sqrt{2\pi}} \exp\left( -\frac{(\ln x - \mu)^2}{2\sigma^2} \right) \qquad x > 0 \tag{A1}
$$

and the density proportional to $x f(x)$ is given by $g(x) = 1/A \cdot x f(x)$, where $A$ is a normalizing constant such that $g(x)$ is a
density. Denote by $G$ the distribution with density $g$. We approximate the median of $G$ by numerically integrating $g(x)$ from
0 to $m$ in R and conducting a grid search for the closest value of $m$ such that $g(m) \cong 0.5$ on a grid with spacing 0.01. The

optimal predictor for the absolute percent error is the functional $\mathrm{med}^{(-1)}(F)$–that is, the median of the distribution with density
proportional to $f(x)/x$–and is found with the same approximation method.

## Appendix B:  Hyperparameter tuning for XGBoost models within the IIS algorithm

XGBoost is used twice in this study: once as the underlying model in the Iterative Input Selection (IIS) algorithm and once
as a predictive performance benchmark in Section 5.1. The two applications are very different and require different hyper-

parameters. In both cases, suitable hyperparameters were chosen by grid-search and cross validation. Here we report the
hyperparameter optimization set up used in the XGBoost models for the IIS algorithm

We used squared error loss as the objective function and tuned the following hyperparameters on the indicated ranges: the
percentage of observations subsampled at each boosting step (0.1-1); the minimum number of instances needed in each node
(1-7); and the shrinkage parameter $\eta$ (0.01-0.1). The number of boosting iterations was evaluated up to a maximum number of

999 iterations. For the XGBoost models used within the IIS algorithm, tree depth was fixed at 1. Hyperparameter tuning was
conducted on a grid search within a 10-fold cross-validation scheme using all possible parameter combinations and an early
stopping criterion for the number of boosting iterations, where the algorithm stopped after 25 rounds without improvement in
the error rate as determined by a chosen evaluation metric. The ranges of the hyperparameters were chosen based on experience
with the data set and recommended XGBoost practices. Hyperparameters were optimized separately for the 1 hour and 24 hour

durations. The evaluation metric used in hyperparameter tuning cross validation is the MAE.

## Appendix C:  Details of the machine learning based pre-selection step

Use of XGBoost within the modular structure of the IIS algorithm was first proposed by Alsahaf et al. (2022). The primary
benefit to this is that use of a boosted tree ensemble—rather than a bagged tree ensemble such as the Extra-Trees routine



originally proposed in Galelli and Castelletti (2013)—solves the issue of significance splitting in the input ranking algorithm.
Significance splitting is when the importance scores of two or more redundant variables are split evenly. This can occur when the tree ensemble is subsampled or bootstrapped, as in bagging and random forest. The algorithm in Galelli and Castelletti (2013) accounts for this by including a secondary evaluation step of the variable ranking to reduce the impact of significance splitting. However, the success of this secondary step is reliant on hyperparameter choice (Galelli and Castelletti, 2013). The use of a boosted tree ensemble inherently solves this issue. Selecting XGBoost as the boosted tree ensemble is a natural choice:
it has established use in hydrology (Zounemat-Kermani et al., 2021), is computationally efficient, is available as a package in both R and Python, and has a large and active user base.

Within the IIS algorithm, we use the additive gain as the importance score in the input ranking algorithm. As part of the model-fitting process XGBoost uses a scoring function that takes into account the improvement in the objective function (in this case, mean squared error) resulting from the inclusion of each variable. The additive gain of a variable is the sum of its
gain across all boosting rounds. For details, see Chen et al. (2015). For a more robust approach, we adopt the method proposed in Laimighofer et al. (2022b), where the initial variable ranking is averaged over 25 bootstrap samples. Then the gain of each variable for the final variable ranking is the ratio of the individual additive gain to the total gain over all variables. The hyperparameters are the same both for the input ranking and the model used to test predictive performance of an additional variable.

The automatic stopping condition in IIS requires choice of both a suitable distance metric for measuring predictive accuracy of the chosen variable set and a threshold value above which a change in predictive accuracy between the proposed sets is considered insignificant. We used mean absolute percent error (MAPE) as the distance metric. We set the threshold value to 0.1, meaning we stop selecting new variable sets when the new set results in, on average, a less than 1 l/s/km2 improvement in median flood prediction. The evaluation of the distance metric takes place across a $k$-fold cross validation approach to increase
robustness. The dataset is divided into $k$ mutually exclusive subsets of equal size, and the predictive model is fit $k$ times. In each iteration, the model is validated on one of the $k$ folds and calibrated using the other $k-1$ folds. The predicted accuracy associated with adding a particular feature is estimated as the average value of the chosen metric over the $k$ validations. We used 10-fold validation for our data set.

The main computational burden of the IIS algorithm is in this repeated model fitting required for computation of the distance
metric in the $k$-fold cross validation: if $m$ potential variables are evaluated at each step, $m * k$ models must be fit. Thus the choice of the number of top-ranked variables to evaluate at each step is important for model performance. While the variable ranking at each step is always computed over the entire variable set, the search space (i.e. the number of variables individually evaluated for predictive performance) can be reduced to a user-specified number of variables. We used the top 15 variables.

In this study the IIS algorithm is used within a resampling scheme. We split our data set into ten non-overlapping folds and
repeatedly apply the IIS algorithm while withholding one of the folds at a time. A visual explanation of the IIS algorithm and this resampling scheme is found in Fig. C1.





**Figure C1.** Visual depiction of the variable pre-selection scheme, showing the IIS algorithm and the resampling method.





**Table E1.** Regions are mid-south-west Norway and eastern Norway + Finmark. The "region" model is fit on the subregions, and evaluation metrics are calculated from all stations included in analysis. There were no statistically significant differences between the two models on either duration.

| Duration | Name | Evaluation metric | | | | |
|---|---|---|---|---|---|---|
| | | RMSE [l/s/km2] | CRPS [l/s/km2] | MAE [l/s/km2] | MRE [%] | MAPE [%] |
| 1 hour | floodGAM | 122.2 | 61.2 | 84.9 | 20.4 | 20.5 |
| | floodGAM, regions | 120.5 | 59.1 | 82.8 | 19.6 | 19.2 |
| 24 hours | floodGAM | 84.1 | 42.7 | 59.5 | 17.0 | 17.5 |
| | floodGAM, regions | 87.4 | 43.8 | 61.6 | 16.7 | 17.8 |

## C1 Full grid output

## Appendix D: Supplementary figures for model evaluation metrics

## Appendix E: Regional assessment

The predictive performance of floodGAM was not significantly improved by splitting the area of study (Norway) into hydrologically homogeneous regions and fitting floodGAM within the regions. Two regions were used: mid-, south and west Norway (region 1) and east Norway and Finmark (region 2); see Fig. E1. The regions are those defined in Hegdahl et al. (2019). Within each region, we ran a 10-fold cross validation; predictive performance metrics were calculated between regional model predictions and the hold out data. The performance metrics were then summarized across the whole of Norway so they could be

compared to the metrics from floodGAM fit to the entire country. Note that this means the hold-out data between floodGAM and "floodGAM, regions" is not identical; however, given the similarities between the metrics it is unlikely this variation in the cross-validation is very influential. Table E1 reports the evaluation metrics from both floodGAM fit to the entire country and floodGAM fit within regions. There were no statistically significant differences between the reported evaluation metrics at either duration.

## 680 Appendix F: Seasonal variations in hydro-climatic predictors

To further investigate the relationships found by floodGAM, we compute seasonal maxima for two seasons: a summer season from April-July and a winter season from August-March. The full model—all seven predictors–is fit on both seasons, and the select argument of the gam() function is set to 'True' such that any predictors found to be irrelevant can be shrunk out of the seasonal models. This serves as a check on the significance of the two climate characteristics ($W_{Apr}$ and $P_{Sep}$) for the seasonal

maxima since April is excluded from the winter months, and September is excluded from the summer months. Figure F1 shows the estimated smooth components for the seasonal maxima along with the associated 95 % estimation uncertainty intervals.



The estimated smooth components for the annual maxima are underlaid as dashed grey lines. The location of data points for each predictor is indicated along the x axis, and y axis ranges are duplicated from Fig. 7.

Figure F1 shows that all four geographical catchment descriptors and one of the hydro-climatic cdescriptors–$Q_N$–have a consistent shape across durations and seasons. However, splitting on seasons changes the shape of the smooth components for the other two hydro-climatic characteristics, $W_{Apr}$ and $P_{Sep}$. For the spring season, $P_{Sep}$ is shrunk out of the model entirely whereas the relationship between $W_{Apr}$ and median spring floods is decreasing.

The autumn/winter seasonal maxima, however, find both $W_{Apr}$ and $P_{Sep}$ to be significant. The shapes of the relationships between the catchment descriptors for the annual model is replicated in the autumn/winter season, except for $P_{Sep}$ where a slightly concave relationship is seen.

*Data availability.* The flood and hydrological data were extracted from the National Hydrological Database (Hydra II) hosted by the Norwegian Water Resources and Energy Directorate (NVE). The data used in this analysis are published at https://doi.org/10.5281/zenodo.8415076

*Author contributions.* DMB, KE, TLT developed the concept and TK and CYX supported the analyses. DMB performed the formal analyses, produced the figures, and wrote the first draft of the paper, which was revised by KE, TK, TLT, CYX

*Competing interests.* The authors declare the following financial interests/personal relationships which may be considered as potential competing interests: Kolbjørn Engeland and Chong-Yu Xu report financial support was provided by Research Council of Norway.

*Acknowledgements.* The authors would like to thank Mads-Peter Dahl for help with data selection.

*Financial support.* This work was supported by the Research Council of Norway through grant nr. 302457 "Climate adjusted design values for extreme precipitation and flooding" (ClimDesign) and FRINATEK Project 274310.



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



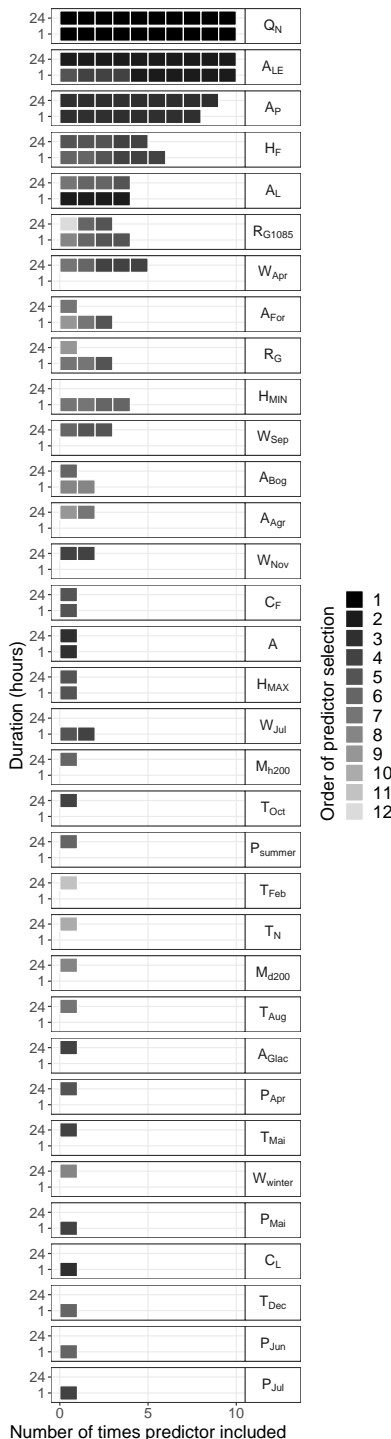

**Figure C2.** Full results from variable pre-selection. The horizontal axis represents the number of times a variable was chosen. The vertical axis indicates the duration that generated the covariate set. The color indicates the order of variable selection within the IIS algorithm. Variables selected first tend to be those that are most informative.





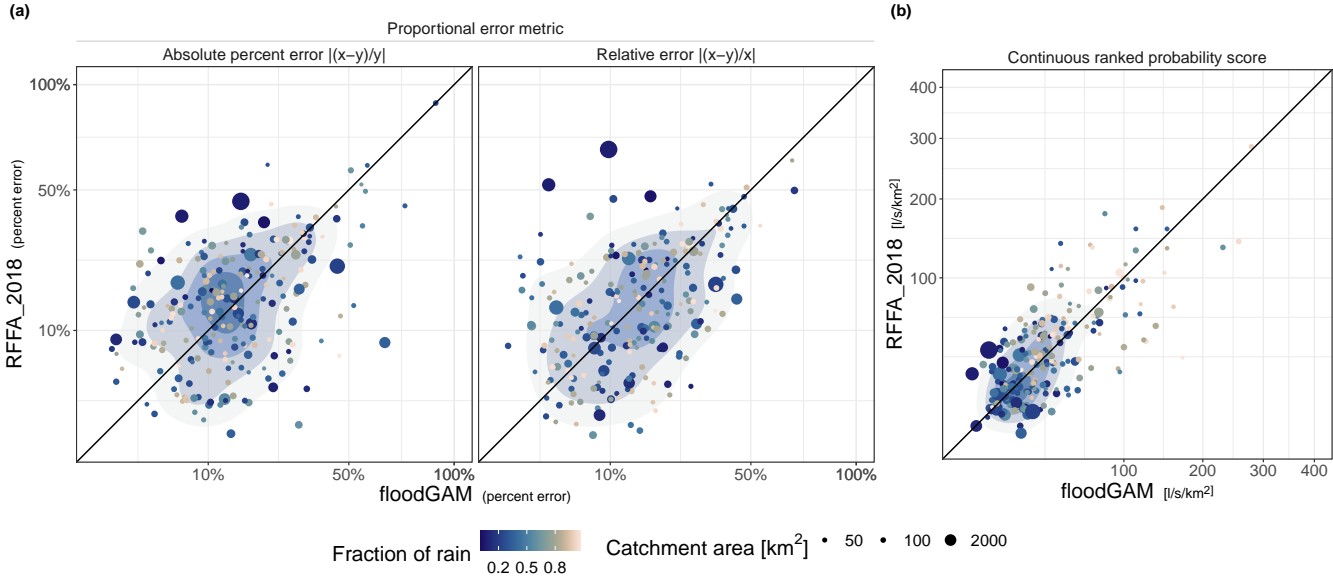

**Figure D1.** Model to model comparison on absolute percent error, relative error, and the continuous ranked probability score for RFFA_2018 and floodGAM on the 24 hour duration. In the panel headers, $x$ represents the predicted value and $y$ the observed value. Points falling above the diagonal line indicate stations where RFFA_2018 performed worse than floodGAM. Points falling below the diagonal line indicate stations where floodGAM performed worse than RFFA_2018. The 2D kernel density estimation of point density is underlaid to aid visual interpretation. Point size shows catchment area, point color indicates the fraction of rain contribution to flood.



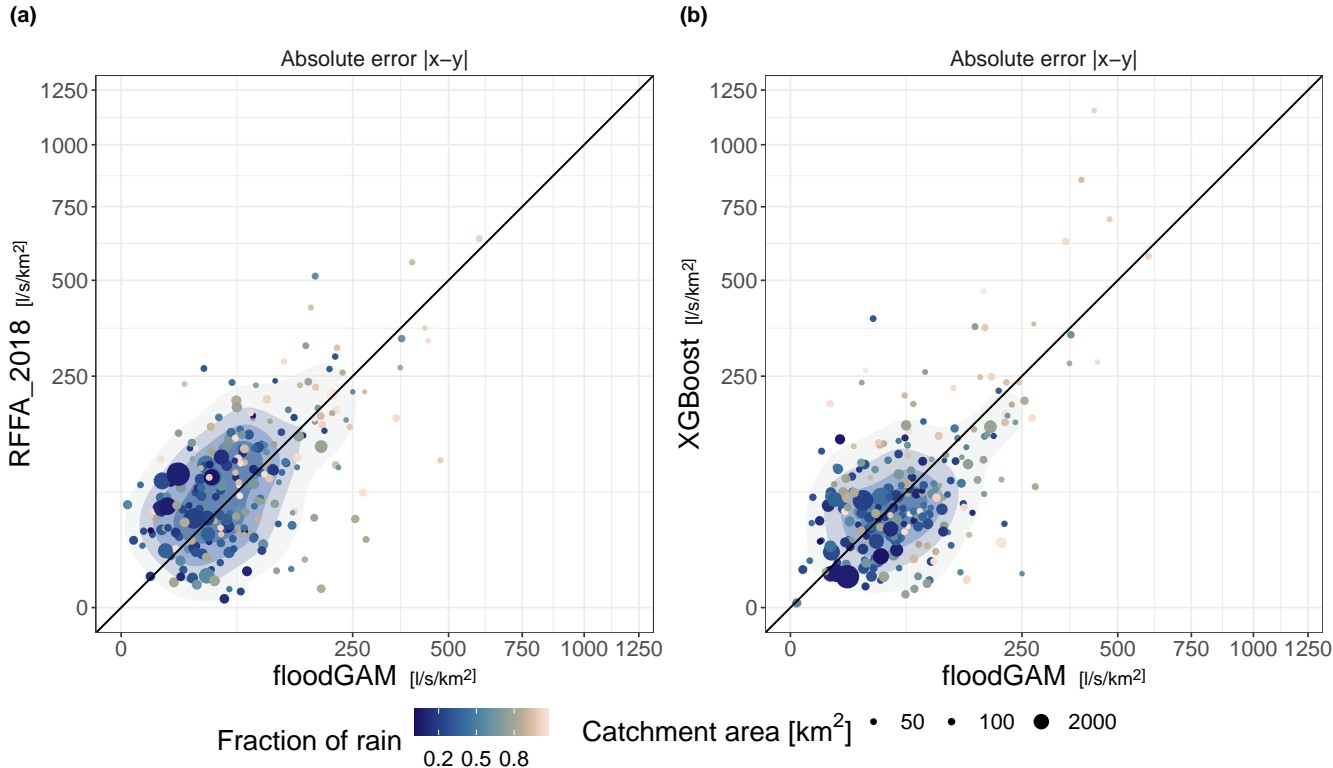

**Figure D2.** Model to model comparison on absolute error for RFFA_2018 vs floodGAM and XGBoost vs floodGAM on the 1 hour duration. In the panel headers, $x$ represents the predicted value and $y$ the observed value. Points falling above the diagonal line indicate stations where the comparative model (RFFA_2018 or XGBoost) performed worse than floodGAM, and vice versa for points falling below the diagonal line. The 2D kernel density estimation of point density is underlaid to aid visual interpretation. Point size shows catchment area, point color indicates the fraction of rain contribution to flood.





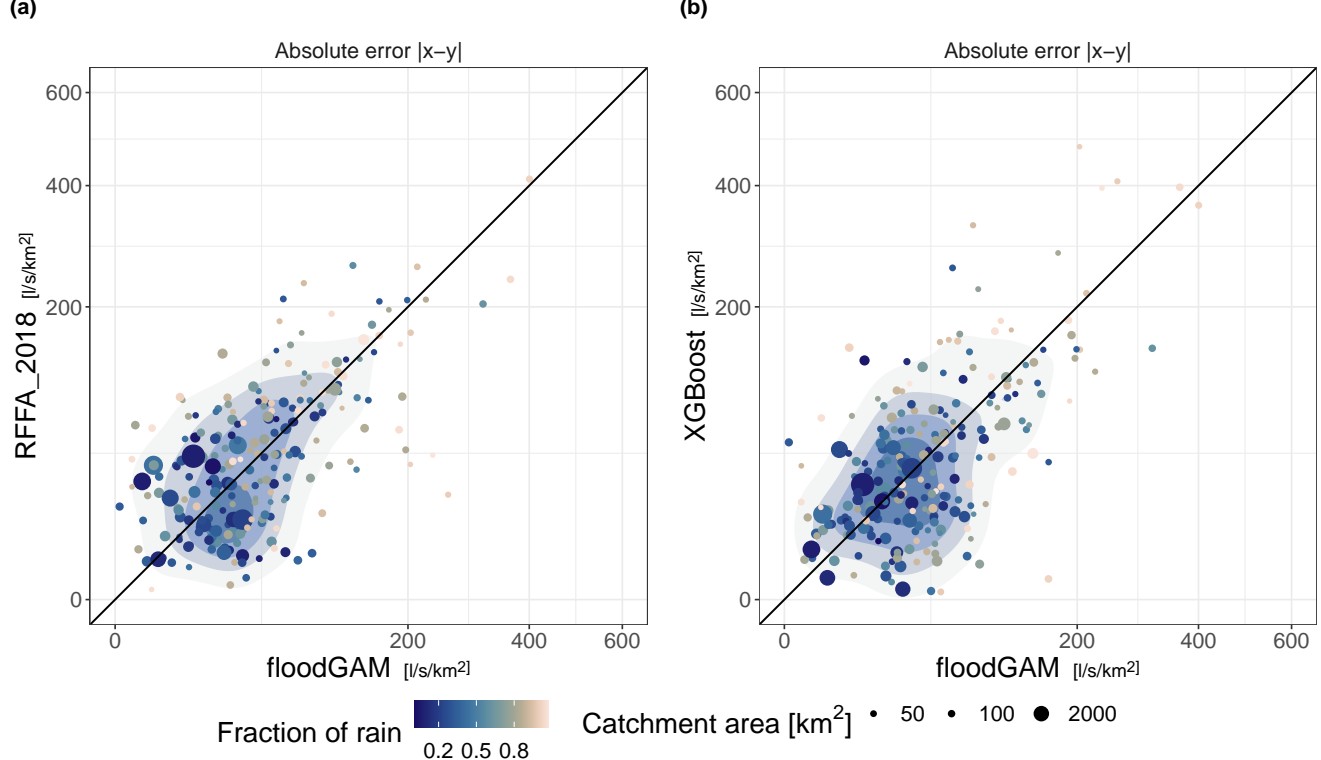

**Figure D3.** Model to model comparison on absolute error for RFFA_2018 vs floodGAM and XGBoost vs floodGAM on the 24 hour duration. In the panel headers, $x$ represents the predicted value and $y$ the observed value. Points falling above the diagonal line indicate stations where the comparative model (RFFA_2018 or XGBoost) performed worse than floodGAM, and vice versa for points falling below the diagonal line. The 2D kernel density estimation of point density is underlaid to aid visual interpretation. Point size shows catchment area, point color indicates the fraction of rain contribution to flood.



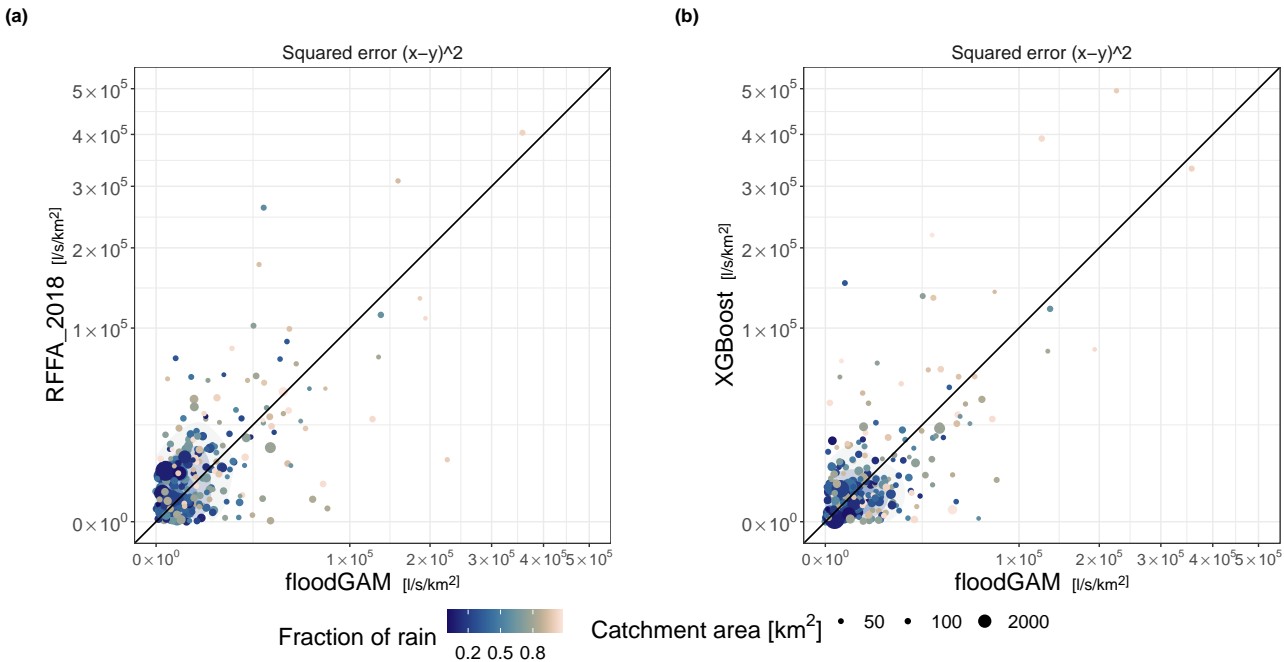

**Figure D4.** Model to model comparison on squared error for RFFA_2018 vs floodGAM and XGBoost vs floodGAM on the 1 hour duration. In the panel headers, $x$ represents the predicted value and $y$ the observed value. Points falling above the diagonal line indicate stations where the comparative model (RFFA_2018 or XGBoost) performed worse than floodGAM, and vice versa for points falling below the diagonal line. The 2D kernel density estimation of point density is underlaid to aid visual interpretation. Point size shows catchment area, point color indicates the fraction of rain contribution to flood.



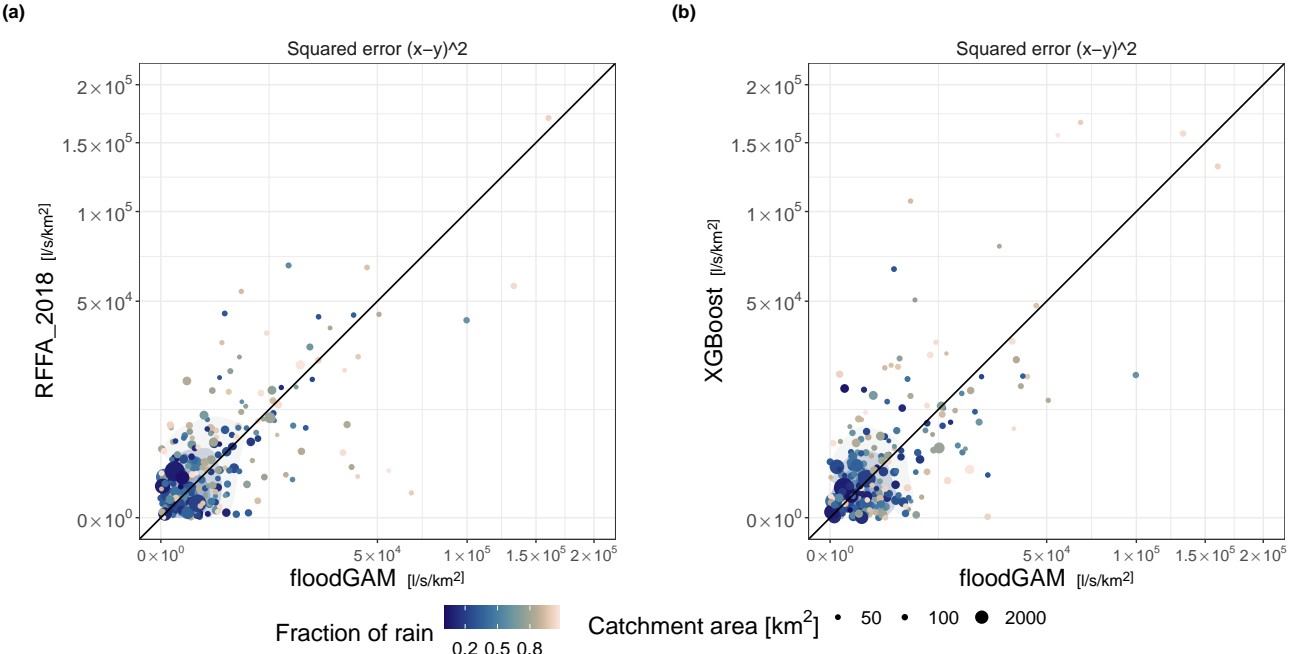

**Figure D5.** Model to model comparison on squared error for RFFA_2018 vs floodGAM and XGBoost vs floodGAM on the 24 hour duration. In the panel headers, $x$ represents the predicted value and $y$ the observed value. Points falling above the diagonal line indicate stations where the comparative model (RFFA_2018 or XGBoost) performed worse than floodGAM, and vice versa for points falling below the diagonal line. The 2D kernel density estimation of point density is underlaid to aid visual interpretation. Point size shows catchment area, point color indicates the fraction of rain contribution to flood.



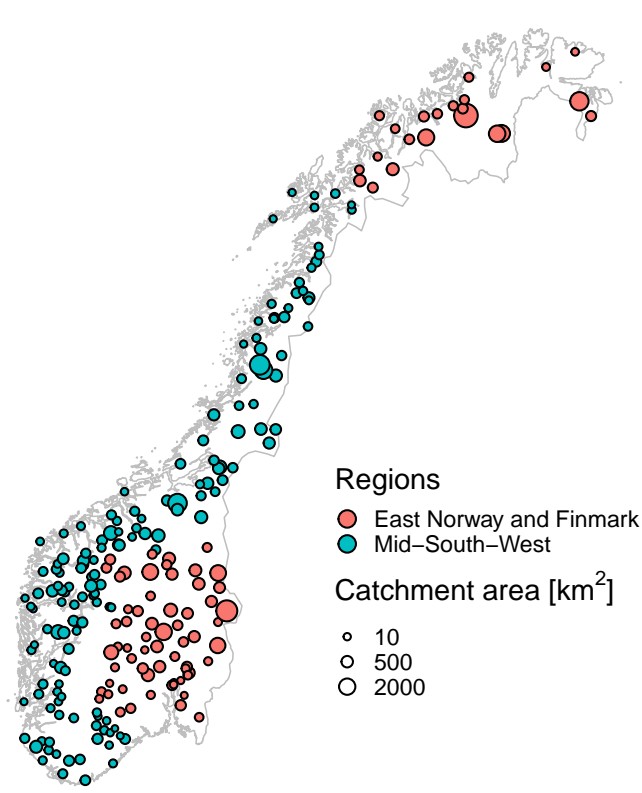

**Figure E1.** Regional groupings for the 232 stations used in this study. Regions are mid-, south and west Norway and east Norway and Finmark; these regions are defined in Hegdahl et al. (2019).



**Figure F1.** Partial response curves by season and duration. The summer season is April-July. The winter season is August-March. 95 % estimation uncertainty intervals for the seasonal smooth components are shown in shading, and smooth components from the annual maxima model are underlaid as dashed or dotted lines. A flat effect means a predictor was selected out of the model by shrinkage. $P_{Sep}$ is selected out of the summer season. The smooth components are shown on the link scale, and units for predictors are shown in panel titles. Location of data points for each predictor are shown as tick marks on the x axis. Y axis ranges span the same magnitude for each panel.