# Peer review of "Regional index flood estimation at multiple durations with generalized additive models"

_EGUsphere, 2023_

## Author Comment (AC1)

**Response to the reviewers**
* * *
**Reviewer #2**

We would like to thank the reviewer for their comments and recognize the time commitment such a review requires. We very much appreciate each of the comments we received. Please find our responses below.

1. ***The title is confusing and misleading in different aspects, such as: - different durations could be understood simultaneously or as a variable - the index flood model is based on quantile while the paper treats only the median (fixed quantile order)***

Reply: We appreciate the input and will change the title to "Regional median flood estimation with generalized additive models: model selection across durations" to avoid confusion. We are happy to consider further revisions if necessary.

2. ***why not using directly and fully Machine Learning? The compatibility between a statistical model GAM and variable selection method based on ML should be discussed. Especially, later in the paper, there is a formal method to select variables for GAM (implemented in mgcv package). In the same idea, (line 43) I'm wondering if the authors are using a modified version of IIS. Hence, this method should be checked and validated before for this choice/context. The question is about the compatibility of this change.***

Reply: We do implement and test a full machine learning model (XGBoost). However, XGBoost does not provide uncertainty estimates. It is also not easily interpreted. We therefore consider the GAM and log-linear model in addition.

The reviewer comments on the availability of a formal method to select GAM predictors within mgcv. Indeed, the predictor set used in the GAM was validated by the formal statistical (shrinkage) methods within the mgcv package (lines 287-288). We apologize that this was not clear in the original version of the manuscript.

Our end goal was always use of the shrinkage methods within mgcv, but our original predictor set (Table 1) contained too many potential variables. Many of these variables are collinear. The shrinkage methods implemented in mgcv are not feasible for such a large, collinear predictor set. In order to use the methods in mgcv, we must select a subset of predictors. Rather than relying solely on expert judgement to select this subset, we use in addition a data-driven approach.

The workflow for predictor selection therefore has three steps. First, a machine learning-based algorithm for predictor selection is applied to the full regional covariate set to generate a "pre-selection set". Next, expert judgement decides if this pre-selection set is reasonable, and adds or subtracts predictors as needed. In our study, we added a precipitation variable ($P_{Sep}$). Finally, the shrinkage methods within mgcv are applied to the chosen subset. In our study, every predictor in the chosen subset was deemed significant by mgcv. This coincidence might be why the predictor selection process seemed somewhat ambiguous.

The idea with this workflow is that the first step has the potential to uncover predictor information that was previously unclear. At the same time, as discussed in Section 3 (study design), the output from the machine learning pre-selection step is kept completely separate from the rest of the analysis, i.e.

predictors proposed by either machine-learning pre-selection or expert judgement are *not* used until they are validated by formal, GAM-specific selection methods that have been rigorously compared to existing methodologies.

Predictor selection for data-driven models (e.g. GAMs) is in general challenging and a major roadblock to setting up analyses such as this one. Hence, we find it beneficial to include our work with the machine-learning based pre-selection approach, even though it serves as just a supporting component to expert judgement, and all formal variable selection is done within the GAM. A machine-learning based pre-selection is in no way a "silver bullet" that is guaranteed to produce a predictor set that will be useful to a GAM. This is discussed on lines 570-577. Development of this pre-selection required a series of careful choices about which algorithm and machine learning model architecture to use, ensuring they could complement GAM development. We discuss and provide rational for the major choices involved. For example, we discuss our choice to use the IIS algorithm to cut down on the collinearity of the selected predictors, and our choice to use a boosted tree ensemble instead of a bagged tree ensemble within the IIS algorithm (Section 4.1, lines 230-259 and Appendix C, lines 631-641). Following the reviewer's comments and suggestions we will provide more explanation on the choices in our procedure and the pros and cons of other alternatives.

3. ***around lines 40-45: this text is ambiguous and not well justified/ motivated. It is based on a unique old paper (see next comments for recent papers). This is part is crucial and motivates the study. Hence, the paper motivation and foundation are questionable.***

Reply: We will revise this section of the introduction to emphasize that the motivation behind evaluating duration-specific differences in regression models comes from the practical necessity of understanding whether a model's performance will decline when applied to durations different from the one it was originally developed for. The publication mentioned was chosen as an illustrative example since it refers to a class of models that simultaneously estimate several durations at once, which of course would be problematic if there were, in fact, duration specific differences.

4. ***Indeed it is more informative to include the duration in the modeling. However, to deal with the duration, it is now appropriate to consider a multivariate framework involving the duration as a variable and simultaneously with other variable like the peak and/volume. The multivariate regional framework, index flood model, is already developed (e.g. Requena et al. 2016, J. of Hydrology; Azam et al. 2018, Water).***

Reply: There are important differences between the current study and the multivariate frameworks mentioned above. We apologize that these differences were not made clear in the original manuscript and will clarify in a revised version of the manuscript.

Regional multivariate frameworks that "treat the duration as a variable" focus on explicitly modeling the dependency structure between different multivariate aspects of events (for example, drought duration and severity [Azam et al., 2018] and flood volume and peak for [Requena et al., 2016]). The modeled dependencies are then transferred to ungauged locations.

Transfer of variable dependencies to ungauged catchments is not a goal of this study. Instead, we build models independently for each duration and examine whether there are statistically significant differences in the relationships between catchment descriptors and the response across different durations. The focus in our study is on this catchment descriptor – response relationship. A goal is to develop a better understanding of the situations where specific regression models may or may

not perform well. This is important groundwork when formulating more complex regionalization approaches, and we hope our study can contribute to this foundational aspect.

Both approaches (the event-based, multivariate approach and the aggregation-based approach we take in this study) are useful, and can be used to find different types of design values where duration is involved. Our design values are meant to support engineering design which requires flood volumes for pre-determined durations, sometimes averaged across several flood events, rather than the multivariate variability of specific flood events.

5. *why this and only these values (1h and 24h)?*

Reply: These are the values that are most relevant to flood guidelines in Norway.

6. ***Around line 60: Dealing with nonlinearity is not only through transformation but directly using nonlinear approaches (see e.g. Ouali et al. 2017, J. Advances in Modeling Earth Systems; Cannon 2018, Stochastic environmental research and risk).***

Reply: We thank the reviewer for this contribution and will update the manuscript to provide this context on nonlinear approaches.

7. *line 108: not sure about this statement, especially no refs given. As far as I know, variable selection is not the strength of ML. I don't know what is reported in Guisan et al. 2002, but it may be not up to date (given the fast development of ML).*

Reply: We will update our references following lines 108-110 to include the more recent publication [Kovács, 2022]; the [Guisan et al., 2002] reference was chosen because it was very clearly written and the idea proposed focused on a broad class of machine learning models (classification and regression tree techniques, or CARTs) that are still in use today. Variable (or feature) selection is a prominent machine learning discipline [Guyon and Elisseeff, 2003] and variable selection by ensemble methods, in particular, has been a focus in recent years [Bolón-Canedo and Alonso-Betanzos, 2019]. The approach used in our study is an ensemble method, i.e. a gradient boosted tree ensemble.

8. *line 115: I'm surprised to see that such an important topic is treated only in the hydrological framework. It is questionable to heavily rely on this.*

Reply: We selected examples from hydrology to demonstrate that these approaches are used in the hydrological community. We will update the manuscript to provide some additional examples from the broader category of applied statistics.

9. ***Last line page 4: This assumption is either strong or in contradiction with the problematic to be treated in the paper.***

Reply: We focus on identification of duration-specific differences in the relationship between predictors and the median flood. We cannot do this if both (i) the predictor set itself and (ii) the data-driven relationship between predictors and response is changing at each duration. Moreover, we investigated if different durations benefited from having different predictor sets, but found no compelling evidence to support this (see Fig. 4).

10. ***It is important to provide an equation for "median annual maximum flood" to be explicit and avoid confusion.***

Reply: We will include an equation for the median annual maximum flood.

11. ***Some parts of the methodology should be in the results section (section 4.2 and from line 250).***

Reply: We realize placing the predictor selection prior to the results is unusual, but we feel the story is more understandable this way, and we want to emphasize the model performance and model interpretation in the results section. We are, however, happy to reconsider if the reviewer feels strongly about this.

12. ***Equation 6: something is missing or wrong. The right-hand side does not depend on i (so the summation is over what?).***

Reply: We thank the reviewer for noticing the indexing variable was omitted here and will fix this in a revised version of the manuscript.

13. ***Using the term permutation test could be misleading since this is a generic term on how to obtain p-value.***

Reply: The term *permutation test* is standard, but the test can also be called a Fisher permutation or randomization test [Fisher, 1936, Good, 2013, Holt and Sullivan, 2023]. We will update our description in the manuscript.

**References**

[Azam et al., 2018] Azam, M., Maeng, S. J., Kim, H. S., and Murtazaev, A. (2018). Copula-based stochastic simulation for regional drought risk assessment in south korea. *Water*, 10(4):359.

[Bolón-Canedo and Alonso-Betanzos, 2019] Bolón-Canedo, V. and Alonso-Betanzos, A. (2019). Ensembles for feature selection: A review and future trends. *Information Fusion*, 52:1–12.

[Fisher, 1936] Fisher, R. A. (1936). Design of experiments. *British Medical Journal*, 1(3923):554.

[Good, 2013] Good, P. (2013). *Permutation tests: a practical guide to resampling methods for testing hypotheses*. Springer Science & Business Media.

[Guisan et al., 2002] Guisan, A., Edwards Jr, T. C., and Hastie, T. (2002). Generalized linear and generalized additive models in studies of species distributions: setting the scene. *Ecological modelling*, 157(2-3):89–100.

[Guyon and Elisseeff, 2003] Guyon, I. and Elisseeff, A. (2003). An introduction to variable and feature selection. *Journal of machine learning research*, 3(Mar):1157–1182.

[Holt and Sullivan, 2023] Holt, C. A. and Sullivan, S. P. (2023). Permutation tests for experimental data. *Experimental Economics*, pages 1–38.

[Kovács, 2022] Kovács, L. (2022). Feature selection algorithms in generalized additive models under concurvity. *Computational Statistics*, pages 1–33.

[Requena et al., 2016] Requena, A. I., Chebana, F., and Mediero, L. (2016). A complete procedure for multivariate index-flood model application. *Journal of Hydrology*, 535:559–580.

---

## Author Comment (AC2)

**Response to the reviewers**
* * *
**1. Reviewer #1**

We would like to thank the reviewer for their time and effort in carefully reviewing our manuscript. We very much appreciate each of the comments we received. Please find our responses below.

**1.1. General comment**

*First, one of the objectives of the study is "prediction of the median flood at ungauged locations" (Line 125-126). I am not quite sure if this is addressed adequately. One of the catchment descriptors is the mean annual runoff of the catchment of the SeNorge 2.0 dataset, which is based on observational data. In my opinion prediction at ungauged locations means, that there is no information about the runoff at this stations, so also no information about the mean annual runoff can be included in a prediction model. I understand that the information maybe necessary for the comparison with the RFFA_2018 model, but I think it can also be beneficial to show that the GAM model performs as good without the mean annual runoff (and leave the mean annual runoff as predictor for the RFFA_2018 model for simplicity).*

Reply: The use of mean annual runoff ("$Q_N$") as a predictor is a modeling choice specific to Norway. The mean annual runoff variable is computed using a gridded hydrological model covering all Norway, making it accessible at both gauged and ungauged locations. The output from the gridded hydrological model is bias-corrected using observed mean annual runoff at selected locations. The reason we use mean annual runoff, instead of alternatives like mean annual precipitation or extreme precipitation, is because in Norway, we possess a more comprehensive spatial coverage of runoff observations–especially at high altitudes–compared to precipitation. The flood frequency guidelines for Norway therefore recommend using modeled runoff [Engeland et al., 2020, Sælthun et al., 1997].

Of course, the uncertainty and bias in $Q_N$ might vary a lot. In some locations, you might use a $Q_N$ that is close to the observed one, whereas at other locations, $Q_N$ might be interpolated. In the first case the estimated error metrics may be a little optimistic. However, for the model-to-model comparison that is a focus of the study, the inclusion of $Q_N$ is sufficient (and necessary) given that regional models in Norway have included $Q_N$ for the past 20 years.

We apologize that this wasn't clear in the original version of the manuscript and will update the discussion to acknowledge this point. Since the difference between the modelled runoff and meterological data in Norway is a question of data quality, it is difficult to construct a fair comparison between models with and without $Q_N$. We experimented with replacing mean annual runoff with a precipitation variable and found reduced performance with the precipitation variable.

**1.2. General comment**

*My second point concerning the prediction in ungauged locations, is the variable selection. If I understood it correctly, the variable selection is performed on the full dataset (with a cross validation scheme), and the selected variables are then used as input for the validation study (again with a cross-validation scheme). If this is correct, the variables are selected on the full dataset, not on a subset, so the prediction error is somehow biased, as the variable selection already included information about the full dataset. If my*

*understanding of the validation scheme is wrong, I would suggest to make this clearer in the methods section.*

Reply:  Thank you for this thoughtful feedback. The variables are, in fact, selected on subsets of data within the cross validation scheme. We thank the reviewer for an opportunity to clarify a key point of the pre-selection and will update the manuscript to explain how the cross-validation is used in the predictor selection / validation schemes. To briefly summarize:

We employ the same 10-fold cross-validation framework for both predictor selection and model validation. First, machine learning-based pre-selection (Section 4.1) is applied to the same 10 folds used in the model validation procedure, with results depicted in Fig. 4. Next, expert judgment interprets Fig. 4 to identify the "potential predictor set," listed in the manuscript as the seven catchment descriptors on lines 279-283. This potential predictor set undergoes formal statistical treatment (selection via shrinkage estimation) within the same 10 folds used in the model validation procedure. Notably, each fold has the potential to yield different predictor selections from the identified set. It was not clear in the original manuscript that we were using the cross validation folds in the shrinkage estimation step of the predictor selection. In our study, shrinkage estimation identified the same predictors for all 10 folds. This, combined with ambiguous language in the original manuscript, made the use of the cross-validation scheme unclear. We will add clarification to Line 289.

**1.3.  General comment**

*The manuscript is a bit too long. Some parts of the methods are in the Appendix, which is fine, but makes it hard to read at some point. Two examples: (i) The description in the Introduction between line 29-46 may be shortened, (ii) or the very detailed description of the response variable (Sect 2./2.1) can may be written more concise.*

Reply:  We agree with the reviewer and will make all efforts to shorten the manuscript in a revised version. We thank the reviewer for the concrete suggestions on how we might do that.

**Additional minor comments**

The additional minor comments provided by reviewer 1 are specific and well thought out. We will incorporate each one into a revised version of the paper. The comment pertaining to line 384 is more substantial than the others so we address it below:

**1.4.  Line 384**

*"XGBoost is assessed only on the MAE; optimal predictors for the other four error metrics are not accessible for XGBoost when the data are assumed log normal." Why is XGBoost then used as for the variable selection procedure - and the MAPE as error metric, if this will produce unreliable result? Additionally, was it considered to alter the loss function in XGBoost for better comparison in terms of the chosen error metrics? I think this should be clarified in the main manuscript.*

Reply:  It is common practice to evaluate predictive performance of competing forecasting methods by assessing their accuracy under a variety of error metrics. As reviewed in [Gneiting, 2011], the various error metrics we use require different point predictions from a predictive distribution to minimize the expected error of a method. However, with forecasting methods such as XGBoost that only output a single best prediction and not the full

predictive distribution, it is not possible to obtain all of the point predictions needed for these metrics in a single run. Furthermore, since changing the loss function might alter the model estimates, treating multiple runs under different loss functions as a single forecast is not appropriate.

However, XGBoost has proven to be a powerful prediction method when a large number of potential—and potentially co-linear— predictors are available. Furthermore, many prediction settings don't require uncertainty assessments, and for such settings, XGBoost has proven to be a popular and powerful method. While we believe that uncertainty assessments are highly relevant in our context, we face the former issue of having many co-linear predictors to choose from, and we find that XGBoost performs well in this aspect of our problem, specifically in selecting a small subset of important predictors. For the sake of completeness, we thus also decided to include the model comparison in the prediction setting to the extent possible, since it also outputs predictions. We see that these nuances have not been sufficiently discussed in the current version of the manuscript, and we will update it accordingly.

The reference to MAPE on line 652 was an error carried over from an earlier version of the manuscript and we thank the reviewer for catching it. The automatic stopping condition relies on MAE, not MAPE

**References**

[Engeland et al., 2020] Engeland, K., Glad, P., Hamududu, B. H., Li, H., Reitan, T., and Stenius, S. M. (2020). Lokal og regional flomfrekvensanalyse. Technical report, NVE.

[Gneiting, 2011] Gneiting, T. (2011). Making and evaluating point forecasts. *Journal of the American Statistical Association*, 106(494):746–762.

[Sælthun et al., 1997] Sælthun, N. R., Tveito, O., Bønsnes, T., and Roald, L. (1997). Regional flomfrekvensanalyse for norske vassdrag. *NVE rapport*, 14:1997.